# IL-33-ST2 axis regulates myeloid cell differentiation and activation enabling effective club cell regeneration

Rania Dagher[1,2,3,4], Alan M. Copenhaver [4], Valerie Besnard[1,2,3], Aaron Berlin [4], Fatima Hamidi[1,2,3], Marielle Maret[1,2,3], Jingya Wang[5], Xiaotao Qu[6], Yashaswi Shrestha [7], Jincheng Wu[6], Gregory Gautier[2,3,8], Rajiv Raja [7], Michel Aubier[1,2,3], Roland Kolbeck[4], Alison A. Humbles [4,9✉] & Marina Pretolani[1,2,3,9✉]

Evidence points to an indispensable function of macrophages in tissue regeneration, yet the underlying molecular mechanisms remain elusive. Here we demonstrate a protective function for the IL-33-ST2 axis in bronchial epithelial repair, and implicate ST2 in myeloid cell differentiation. ST2 deficiency in mice leads to reduced lung myeloid cell infiltration, abnormal alternatively activated macrophage (AAM) function, and impaired epithelial repair post naphthalene-induced injury. Reconstitution of wild type (WT) AAMs to ST2-deficient mice completely restores bronchial re-epithelialization. Central to this mechanism is the direct effect of IL-33-ST2 signaling on monocyte/macrophage differentiation, self-renewal and repairing ability, as evidenced by the downregulation of key pathways regulating myeloid cell cycle, maturation and regenerative function of the epithelial niche in ST2$^{-/-}$ mice. Thus, the IL-33-ST2 axis controls epithelial niche regeneration by activating a large multi-cellular circuit, including monocyte differentiation into competent repairing AAMs, as well as group-2 innate lymphoid cell (ILC2)-mediated AAM activation.

---

[1] Inserm UMR1152—Physiopathologie et Epidémiologie des Maladies Respiratoires, Université Paris, Paris, France. [2] Faculté de Médecine, Université Paris, Paris, France. [3] Laboratoire d'Excellence (LabEx) INFLAMEX and Département Hospitalo-Universitaire (DHU) FIRE, Paris, France. [4] Bioscience COPD/IPF, Research and Early Development, Respiratory & Immunology, BioPharmaceuticals R&D, AstraZeneca, Gaithersburg, MD, USA. [5] Translational Science and Experimental Medicine, Research and Early Development, Respiratory & Immunology, BioPharmaceuticals R&D, AstraZeneca, Gaithersburg, MD, USA. [6] Data Sciences and AI, BioPharmaceuticals R&D, AstraZeneca, Gaithersburg, MD, USA. [7] Translational Medicine, Oncology R&D, AstraZeneca, Gaithersburg, MD, USA. [8] Inserm UMR1149 - Centre de Recherche sur l'Inflammation, Université Paris, Paris, France. [9] These authors contributed equally: Alison A. Humbles, Marina Pretolani. ✉email: alisonhumbles@gmail.com; marina.pretolani@inserm.fr

Normal repair of the lung epithelium involves sequential and differentially regulated phases, including inflammation, proliferation/regeneration, and remodeling[1]. Epithelial regeneration is largely driven by multipotent progenitor stem cells of the bronchial conducting airways, specifically a subset of club progenitor stem cells expressing low levels of club cell secretory protein (CCSP$^{low}$). In response to injury, these cells proliferate and differentiate in order to restore epithelial integrity[2].

Recent evidence has highlighted a central role for macrophages in the repair process of various tissues[3,4] whereby these cells can exhibit remarkable plasticity, allowing them to control immune homeostasis and remodeling by promoting cell proliferation and extracellular matrix (ECM) synthesis[5,6]. Lung resident macrophages express CD206, a marker of AAMs[7] and are thought to originate from three distinct precursors[8,9]. After lung injury, alveolar macrophages are mobilized to repopulate the airways and support tissue repair. This repopulation involves in situ macrophage proliferation, and/or the activation of macrophage precursors, specifically the bone marrow monocytes, which replenish the alveolar space[10]. Thus, depending on the mode and severity of injury, repopulating alveolar macrophages adapt to environmental challenges to accommodate the needs of the lung tissue[11]. Studies conducted in human skin after hematopoietic stem cell transplantation provide indirect evidence that tissue macrophages are long-lived cells and may exist independently of circulating progenitors[12]. Further, in response to different stimuli, human macrophages acquire a diverse spectrum of activation states that correlate to functionally distinct phenotypes in mice[13].

IL-33 is an epithelial-derived cytokine, released during mechanical stress and cellular damage[14–16], which signals via the ST2 receptor[17] and is expressed by several cell types in the lung, including ILC2s and macrophages[15,17,18]. The IL-33-ST2 pathway can induce macrophage alternative activation[19,20], but its direct role in macrophage fate and survival has not been explored.

Here, we identify a paradoxical and protective role for ST2$^+$ lung resident macrophages in club cell regeneration, which requires ILC2 activation and downstream myeloid cell activity within the epithelial niche. We also uncover an additional function for ST2 in myeloid cell differentiation and maturation, which is critical for effective epithelial repair. This study provides insight into how epithelial repair may occur in patients with chronic respiratory disease and enables the design of macrophage-targeted therapies.

## Results

### Three myeloid populations expand in the airway after injury.
The role of macrophages in epithelial repair was investigated using the naphthalene (NA)-induced bronchiolar epithelial injury model[21], which is characterized by a rapid loss of club cells as shown by the decrease of CCSP expression in tissue sections (Fig. 1a, b) and of the mRNA encoding secretoglobin family 1A member 1 (Scgb1a1) in lung homogenates (Supplementary Fig. 1a). Epithelial regeneration was determined by the recovery of CCSP protein and Scgb1a1 mRNA expression, beginning after day 6 (d6, maximal proliferation of club cells), and returning to normal levels by d35 (Fig. 1a, b, Supplementary Fig. 1a). During this epithelial repair phase (d6–d35), we observed an accumulation of F4/80$^+$ myeloid-derived macrophages near to the injured bronchial epithelium (brown staining, Fig. 1c). Total monocytes/macrophages in the bronchoalveolar lavage (BAL) also increased and peaked between d6 and d9, after which numbers declined to baseline (Fig. 1d). Macrophage expansion was associated with an early (d1–d3) increase in BAL fluid levels of IL-1α, IL-13, CCL2, and CXCL10[22], important regulators of monocyte/macrophage function (Fig. 1e).

The majority of cells within the airway lumen were myeloid-derived monocyte and macrophage populations, (Fig. 1f, Supplementary Fig. 1b). We further defined three distinct myeloid cell populations (Fig. 1f, g), namely F4/80$^{low}$ CD11b$^+$ Ly6C$^+$ infiltrating inflammatory monocytes (P1 subset); F4/80$^{int}$ CD11b$^+$ monocyte-derived macrophages (P2 subset); F4/80$^{high}$ SiglecF$^+$ CD11c$^+$ resident alveolar macrophages (P3 subset), and F4/80$^{low}$ annexin V$^+$ apoptotic macrophages (P4 subset) (Supplementary Fig. 1c). The absolute numbers of P1–P3 subsets peaked between d6 and 15 post NA (Fig. 1g), whereas 40% of F4/80$^{low}$ CD11b$^-$ annexin V$^+$ apoptotic macrophages (P4 gate) were detected on d3 but declined by d9 (Fig. 1f). Further phenotyping found that P3 subset exhibited an enhanced AAM phenotype, when compared to P1 and P2 cells and expressed CD206, FIZZ-1, Arg-1, and YM1 (Fig. 1h, Supplementary Fig. 1d), and their counts increased approximately fourfold between d6 and d15 post-NA, when compared to naïve mice (Supplementary Fig. 1d, e). In addition, increased mRNA expression of the AAM markers, Itgax, Cd206, Retnla, Arg1, and Chil3 was observed in total BAL cells isolated from NA-treated mice, when compared to naïve (Supplementary Fig. 1f). NA-induced injury also triggered local macrophage proliferation between d3 and d21 (Fig. 1i), as evidenced by Ki-67$^+$ P2 and P3 subsets (Supplementary Fig. 1g). Proliferation of F4/80$^+$ macrophages was further confirmed by co-immunofluorescence (Supplementary Fig. 1h). Thus, macrophage expansion during epithelial repair involves a significant proliferation of monocyte-derived macrophages (P2) and resident alveolar macrophages (P3), as well as the recruitment of inflammatory monocytes (P1).

### Epithelial regeneration requires resident lung macrophages.
To determine the contribution of alveolar macrophages to bronchial repair, myeloid cells were depleted by administering Clodronate (CL)-containing liposomes[23] to NA-treated mice on days 2, 5, and 8 (Fig. 2a). CL-treated mice exhibited an incomplete bronchial re-epithelialization (Fig. 2b, c), that was associated with ~90% reduction of BAL macrophages, when compared to vehicle containing liposomes, including reduced P2 and P3 numbers, while P1 counts were unaffected by CL treatment (Fig. 2d). In addition, a significant reduction in the expression of AAM markers was seen (Supplementary Fig. 2a, b). Adoptive transfer experiments were performed to decipher which macrophage subset contributed to repair. Given their significant increase during the repair phase (Fig. 1g), P3's (i.e., F480$^{high}$ CD11b$^-$ CD11c$^+$ CD206$^+$) were sorted to ≥98% purity from the airways of NA-injected mice, and transferred into recipient, macrophage-depleted, NA-treated animals, 72 h after the final CL administration (d11, Fig. 2a, Supplementary Fig. 2c). Reconstitution of P3 resident macrophages completely rescued repair in macrophage-depleted mice, (Fig. 2b, c), suggesting that these cells are required for club cell regeneration. We hypothesized that P2's would also contribute to repair by their ability to differentiate into mature P3 cells[9,24,25], therefore, GFP$^+$ P2 cells (F4/80$^{int}$ CD11b$^+$ cells), were transferred as described above, to recipient macrophage-depleted mice and tracked for their surface markers post-NA injury (Fig. 2a). Interestingly, immature, GFP$^+$ monocyte-derived macrophages acquired markers of P3 resident macrophages, namely CD11c and SiglecF, within 1 week of transfer to recipient mice; these maturation markers were still evident 2 weeks post injury (Fig. 1e). Thus, P2's are a transient population which upon injury can readily switch into P3 cells and contribute to replenishment of the P3 resident pool enabling continued repair. This observation is consistent with bleomycin-induced lung injury where 50% of alveolar macrophages were found to be monocyte-derived after fibrosis had resolved[26], in addition to adoptive cell

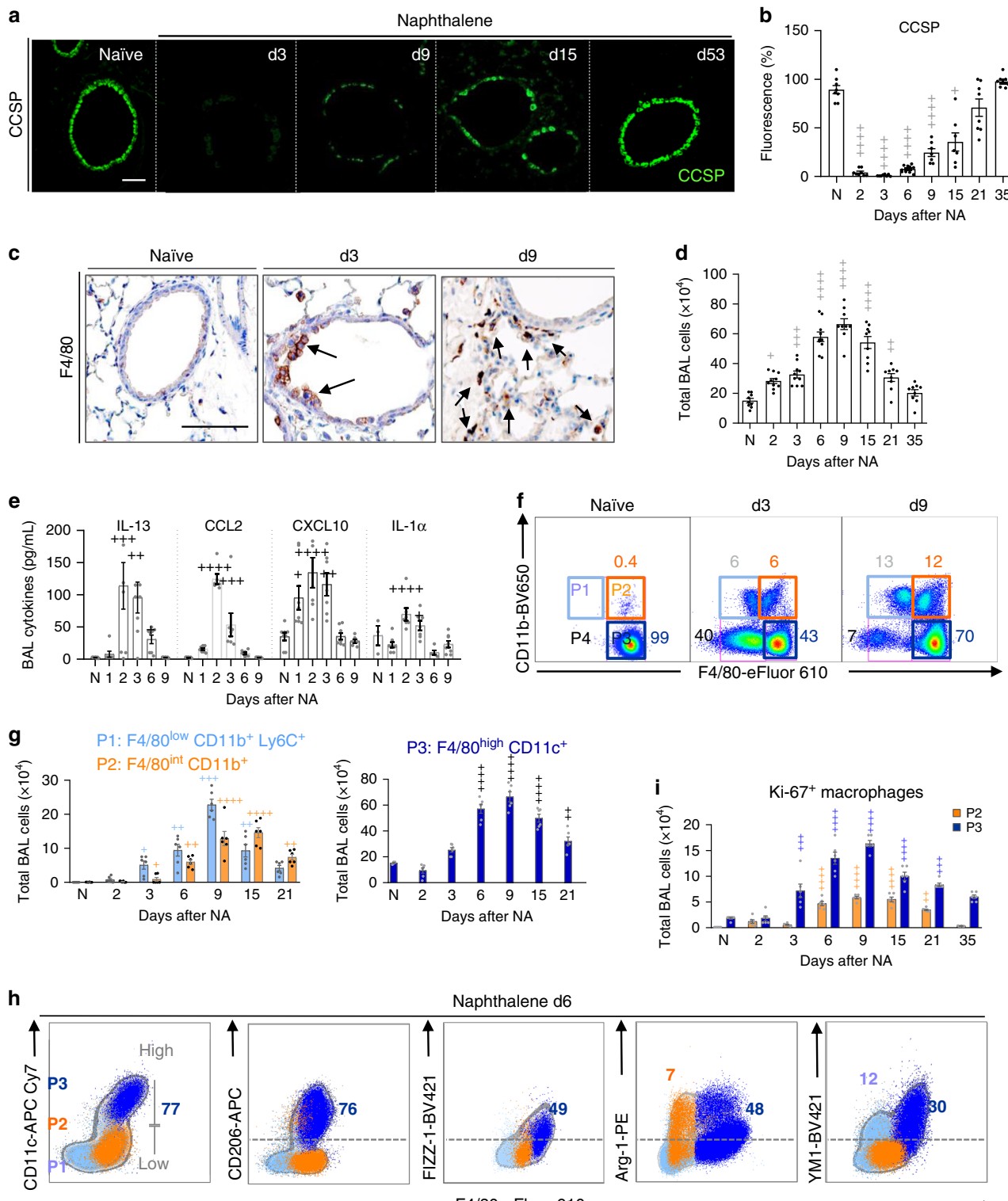

transfer studies demonstrating that CD11c+ alveolar macrophages originate from blood monocytes in response to conditional macrophage depletion[24].

**RNASeq confirms club cell regeneration pathways in P3 cells**. RNAseq analysis was performed on flow-sorted P1–P3 subsets 6d post-NA treatment. Pairwise comparisons between the different subsets revealed 3751 differentially expressed genes (DEG) during monocyte differentiation (P2 vs. P1), while there were 6244 DEG

between P2 and P3 subsets (Supplementary Fig. 3a, Supplementary Datas 1–4). Thus, based on their transcriptome, resident (P3) and recruited macrophages (P2) are functionally distinct. Computational analyses were applied to decipher key functional differences between these myeloid populations. Principal component analysis (PCA) identified transcriptomic clustering of the distinct myeloid subsets and highlighted the similarities between naïve and post injury P3 subsets (Fig. 3a). Additional hierarchical clustering identified three distinct expression patterns

**Fig. 1 Macrophages predominate during epithelial repair and exhibit AAM phenotype. a–i** WT C57BL/6 mice were untreated (naïve, N) or treated with naphthalene (NA) and analyzed at various days thereafter. **a** Bronchiolar epithelium regeneration after NA-induced injury, as assessed by immunofluorescence staining of CCSP in lung tissue sections. **b** Quantification of CCSP expression in lung tissue sections from naïve and NA-treated mice, expressed as percentage of fluorescence within bronchioles (150–400 µm diameter), at the indicated time-points after NA. **c** Immunohistochemical analysis of F4/80 expression (brown deposit) illustrating macrophage localization (black arrows) around the injured bronchiolar epithelium in lung tissue sections. **d** Quantification of the total number of cells in the bronchoalveolar lavage (BAL). **e** Levels of IL-13, CCL2, CXCL10, and IL-1α in BAL supernatants. **f** Monocyte/macrophage subsets (P1–P4). Inflammatory monocytes F4/80$^{low}$ CD11b$^+$ (P1), recruited macrophage F4/80$^{int}$ CD11b$^+$ (P2), resident macrophages F4/80$^{high}$ CD11b$^-$ (P3) and apoptotic macrophages Annexin V$^+$ F4/80$^{low}$ CD11b$^-$ (P4) in the BAL are defined by their gates in (**f**). **g** Total cell numbers of P1–P3 subsets at the indicated time-points after NA administration. **h** Representative FACS profiles of BAL cells obtained on d6 after NA, illustrating the expression of CD206, FIZZ-1, YM1, and Arg-1 in P1–P3 BAL cell subsets, respectively. **i** Quantification of BAL macrophage proliferation as assessed by FACS analysis on P2 and P3 subsets, using Ki-67 staining. Data are from 8 (**a–e**) and 6 (**g, i**) mice, obtained in 3 independent experiments, and represented as mean ± SEM. $^*P < 0.05$, $^{**}P < 0.01$, $^{***}P < 0.001$ and $^{****}P < 0.0001$ between NA-treated and naïve WT mice using one-way ANOVA, Bonferroni post-test. Scale bars in **a** and **c** = 100 µm.

within the P1–P3 subsets, named clusters I–III (Fig. 3b, Supplementary Data 5). Notably, cluster I included 374 highly expressed genes in P1's, which were slightly downregulated upon differentiation into P2 cells and further repressed within P3's. Cluster II exhibited an expression profile of 33 genes which were highly upregulated in P2's compared to P1 and P3 subsets. Upon P2's converting into P3 cells, 299 genes associated with alveolar macrophage homeostasis[27] became more prominent in cluster III compared to cluster I and II (Fig. 3b). Pathway enrichment analysis was performed to understand the biological processes associated with each gene cluster. Genes upregulated during the P2–P3 cellular switching were associated with efferocytosis, cell cycle, biosynthetic processes, secretion of ECM components and growth factors, pathways all involved in repair. In contrast, downregulated genes were mostly associated with development, complement activation, inflammatory responses and signal transduction (Fig. 3b, Supplementary Data 6).

Because epithelial progenitor behavior is regulated by the local environment[28,29], we hypothesized that both recruited and resident macrophages may modulate the club cell niche. The expression of genes associated with epithelial repair including growth factors, ECM remodeling and inflammation between P2 and P3 subsets were compared (Fig. 3c, Supplementary Fig. 3b, Supplementary Data 7). Although P2 cells displayed higher expression of the alveolar regenerating factor *Wnt11*[30], P3 cells were enriched with a larger variety of growth factors required for club cell regeneration, such as *Plet1, Nrg4, Gdf15, Nrp2, Ereg,* and *Mreg*[31–33]. P3's also displayed higher expression of the epithelial niche ECM components, *Ctsk*[34], *Mnt2*[35], and *Krt79*[36] (Fig. 3c, Supplementary Fig. 3b). Interestingly, P2 cells were more profibrogenic and proinflammatory, similar to those reported in lung fibrosis[26], than their P3 counterparts that exhibited more of an immunomodulatory phenotype as seen by the higher expression of *Cd200r*[37] (Fig. 3c). Lastly, we examined the enriched signal transduction pathways within P1 and P2 subsets associated with cluster I in depicting myeloid cell development and differentiation pathways, (i.e., complement activation, innate and immune responses, transcription, cytoskeleton remodeling, and chemotaxis). In addition to the upregulation of IL-4 and IL-13 signaling, we found significant evidence for IL-1 and IL-33 pathway activation, including overexpression of genes encoding *Il1rl1* (ST2) and *Il1racp*, (IL-1RAcP; the co-receptor required for IL-1R1 and ST2 signaling), and subsequent downstream NF-κB activation via the genes encoding the heterodimeric signaling complex MyD88/IRAK1/IRAK2/TRAF6 which activates NF-κB transcription factor[16] (Fig. 3d, e).

**AAM-mediated epithelial repair requires the IL-33-ST2-axis.** Evidence for IL-33 pathway activation is consistent with our

previous report of ST2$^+$ macrophages modulating inflammation during influenza infection[15], thus we examined ST2 function on macrophages during NA-induced repair. Interestingly, *Il1rl1* transcripts were increased in total BAL cells and both ST2$^+$ P2 and ST2$^+$ P3 cells were evident in the lung post-injury (Fig. 4a). Using a commercial monoclonal antibody, we found that ST2 expression appeared to be much higher on P2 subset vs. relatively dim P3 cells (Fig. 4b, left panel). ST2 was not detected on P1 cells (Supplementary Fig. 3c), but ST2 mRNA expression was observed in P1 and P2 subsets 6 days post injury (Fig. 3e). Commercial ST2 antibodies exhibit significant background staining and are best used for highly expressing cells, thus, we attempted to confirm receptor levels using cells from ST2-GFP reporter mice[15,38]. Increased expression of GFP-ST2 was detected on P2 macrophages, (Fig. 4b, right panel), however, the weak fluorescence emission signals from these reporters were not sensitive enough to capture the lower ST2 levels detected on P3 cells by flow cytometry. This observation is consistent with our RNAseq data where *Il1rl1* mRNA levels are high in P2 subset while *Il1racp* mRNA, is relatively low, in contrast, P3 cells have minimal *Il1rl1* mRNA expression but very high *Il1racp* mRNA levels (Fig. 3e). Thus, ST2 expression may be transient during the differentiation of immature P1 cells to mature P3 macrophages following NA-induced injury. This hypothesis is supported by a previous report demonstrating that ST2 expression during endothelial cell differentiation was growth dependent[39].

Consistent with the precedence for IL-33 release during injury[14,15,40,41] and IL-33 expression being reported in subsets of lung epithelial progenitors post infection[14], single cell-RNAseq analysis of mouse bronchial epithelial cells, (Epcam$^+$ Scgb1a1$^+$), demonstrated that a proportion of club cells (33% of Cluster 1, Fig. 4c) constitutively expressed *Il33* mRNA. To confirm club cell expression at homeostasis and decipher whether there was a loss of IL-33 signal following NA-induced desquamation, we optimized an established immunohistological method using a validated antibody[15,42]. Indeed, we showed that club cells expressed IL-33 protein at baseline and a small subset of these were also positive for this cytokine following NA treatment (Fig. 4d, left and middle panel respectively, red/white cells). In addition, we observed a significant reduction in IL-33 immunoreactivity within red-labeled club cells after NA, (Fig. 4d, middle panel; NA-resistant club cells maintain IL-33 expression, white cells d9 post NA; ciliated cells are green), levels of which were restored upon epithelial regeneration (Fig. 4d, right panel, d35 post NA). Since IL-33 is susceptible to oxidation[43] and technically challenging to detect, we investigated the levels of soluble ST2 (sST2), a decoy receptor for IL-33, produced following IL-33 release and thought to predict activation of this pathway[44,45]. Increased amounts of sST2 were found between d1 and d2 post NA while lung IL-33 levels, (i.e., presumably intracellular),

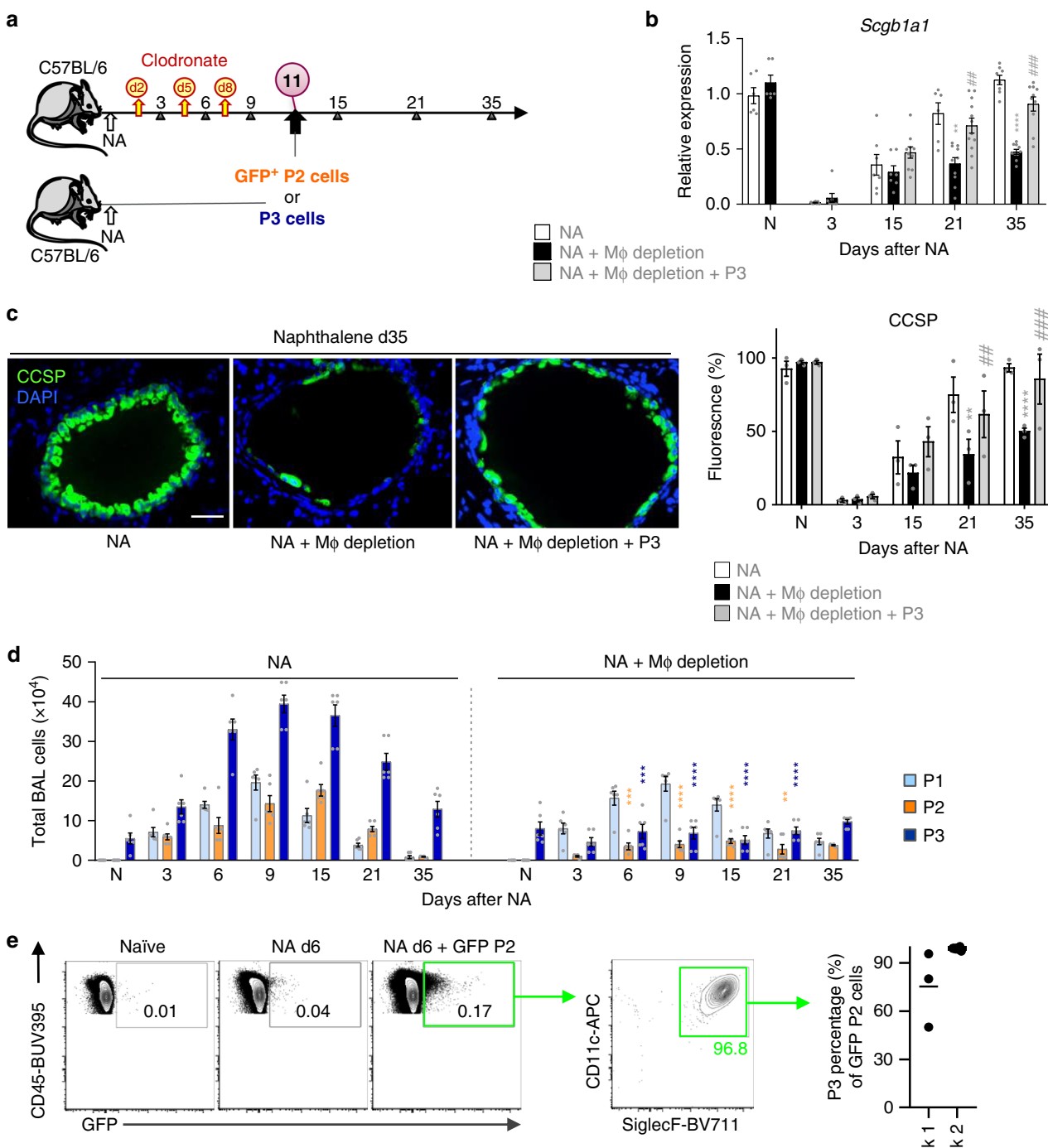

**Fig. 2 AAM resident airway macrophages (P3) are essential for bronchiolar epithelial regeneration. a** Schematic for clodronate (CL)-mediated macrophage depletion during naphthalene treatment (NA + macrophage depletion). NA-treated macrophage-depleted mice were then either given no cells or intratracheally adopted with GFP+ P2 or P3 cells (NA + P3/AAM adoptive transfer). **b** Levels of the *Scgb1a1* mRNA in total lung homogenates from NA-treated mice ± macrophage depletion. **c** Assessment of club cell regeneration after NA-induced injury in lung tissue of NA-treated WT mice (Control) or NA-injected mice treated with clodronate liposomes (Depleted) or AAM adoptively transferred into depleted mice (see Fig. 2a). CCSP immunofluorescence staining was performed at d35. Graph on the right represents CCSP quantification in lung tissue sections, expressed as percentage of fluorescence within bronchioles (150–400 μm diameter). **d** Total BAL cells subdivided into P1–P3 gates as defined in (Fig. 1f) determined by counting and flow cytometry from NA-treated mice ± macrophage depletion. **e** Representative flow plots illustrating the percentage of adoptively transferred GFP+ P2 cells that switched into CD11c+ SiglecF+ cells after 1 week in the lungs of depleted mice. Graph on the right represents the quantification of CD11c+ SiglecF+ GFP+ cells. Data are from 6 to 10 (**b**), 3 (**c**), 6 (**d**), and 3 to 7 (**e**) mice, obtained in three independent experiments and represented as mean ± SEM. $^{*}P < 0.05$, $^{**}P < 0.01$, $^{***}P < 0.001$ and $^{****}P < 0.0001$ between macrophage-depleted and NA-treated WT mice using one-way ANOVA, Bonferroni post-test. $^{\#}P < 0.05$, $^{\#\#}P < 0.01$ and $^{\#\#\#}P < 0.001$ between AAM adoptively transferred and macrophage-depleted WT mice using one-way ANOVA, Bonferroni post-test. Scale bar in **c** = 50 μm.

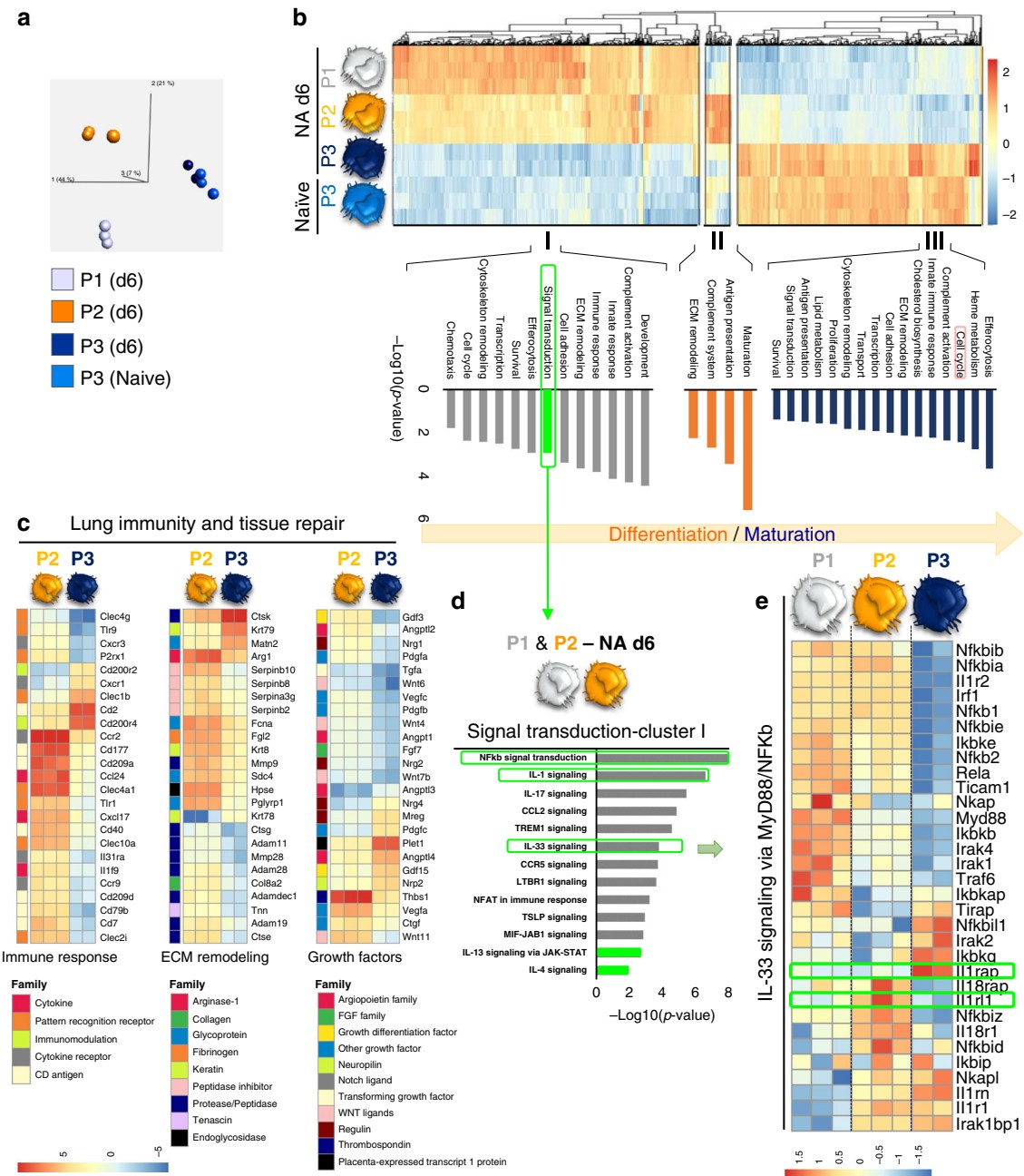

**Fig. 3 Airway macrophages are transcriptionally primed to support epithelial repair and display IL-33-ST2 activation during their differentiation process. a–e** Transcriptional profiling of myeloid cell subsets flow-sorted from naïve and NA-treated mice on d6. **a** Principal component analysis (PCA) of the transcriptomes of flow-sorted monocytes (P1), monocyte-derived macrophages (P2), and resident AAMs (P3) performed on all expressed genes identified from a generalized linear model to perform an ANOVA-like test for differential expression between any conditions in the dataset (FDR step-up procedure q-value < 0.05. **b** k-means clustering of all identified genes revealed core gene signatures specific to P1 (Cluster I), P2 (Cluster II), and P3 (Cluster III). The pathway enriched processes associated with each cluster are shown on the bottom of the heatmap. Scale bar on the bottom denotes relative log$_2$ differences in gene expression for each row. **c** Heatmaps showing the top relative expression of DEG that are associated with macrophage ability to modulate the epithelial cell niche. Those functions include growth factors secretion, extracellular matrix (ECM) remodeling and immune response regulation between P2 and P3 cells. **d, e** Score plot illustrating the signal transduction pathways upregulated in Cluster I (**d**), highlighting the overrepresentation of IL-33 signaling and MyB88/NFkb axis in the different subsets of myeloid cells (**e**).

remained elevated until d6 and d15, respectively (Fig. 4e, f). These data, along with the RNAseq above (Fig. 3d, e), prompted us to examine whether the IL-33-ST2 pathway was required for AAM-mediated epithelial repair. Strikingly, NA-treated mice lacking the ST2 receptor, (Il1rl1$^{-/-}$), exhibited a severe defect in epithelial repair as shown by significantly lower CCSP expression when compared to wild-type (WT) mice (Fig. 5a, b, Supplementary

Fig. 4a). We questioned whether this incomplete re-epithelialization resulted from an impaired differentiation and/or proliferation of club cells. The proportion of CCSP$^{high}$ (terminally differentiated) and CCSP$^{low}$ (potential to proliferate) and Ki-67$^+$CCSP$^{low}$ cells within each bronchiole were quantified by immunofluorescence. In WT mice, NA injury induced the proliferation of NA-resistant club cells (Ki-67$^+$CCSP$^{low}$) that

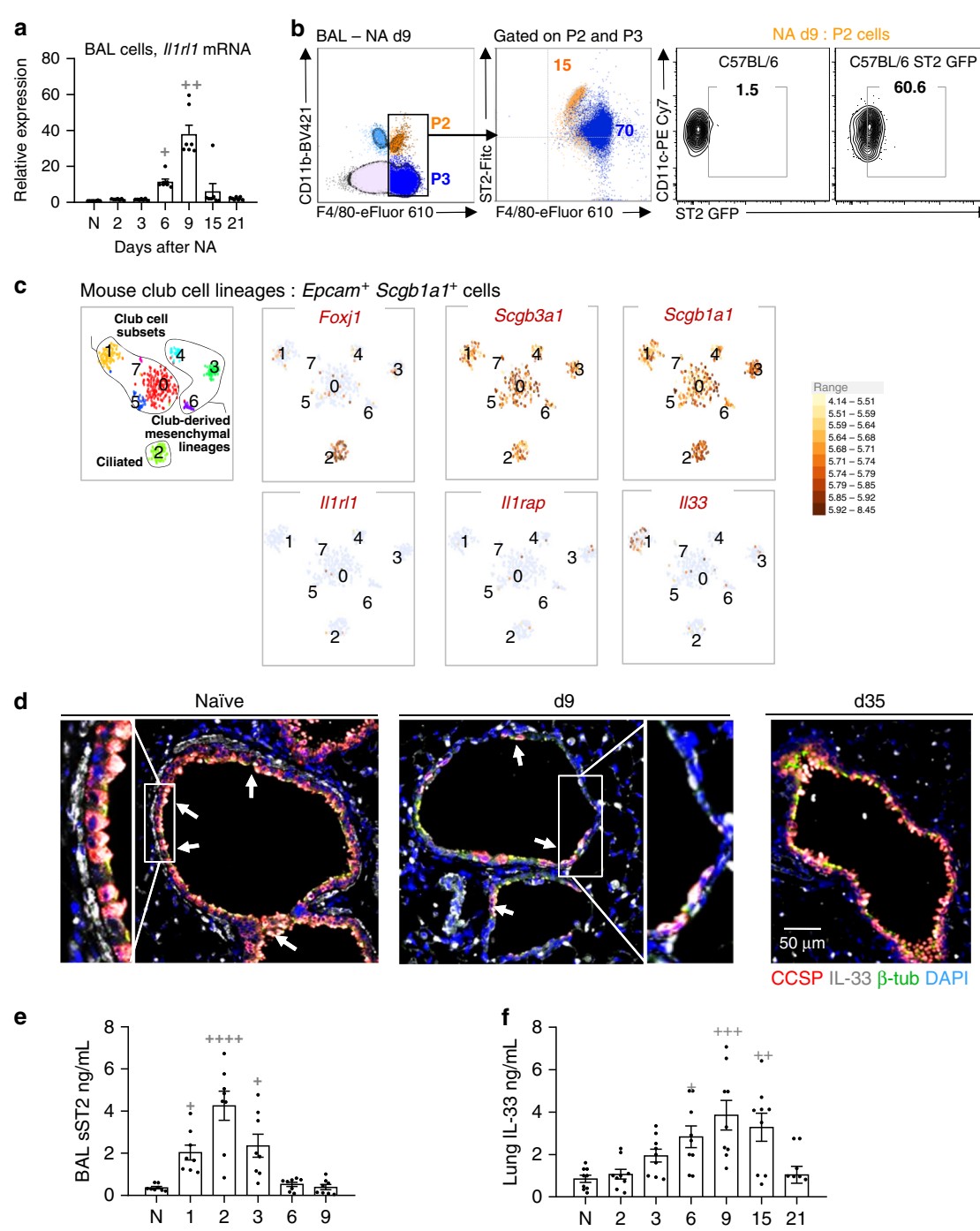

**Fig. 4 IL-33-ST2 axis is activated in airway macrophages during bronchial re-epithelialization. a** Relative mRNA expression of *Il1rl1* as determined by qPCR for total BAL macrophages. **b** FACS quantification as percentage of ST2 expression on P2 and P3 cells of total cells in BAL. Results were confirmed using GFP ST2 reporter mice in the right panel. **c** Single-cell RNA-sequencing analysis of mouse naïve bronchial epithelial lineages showing t-SNE plots of club cell lineages as identified by *Scgb1a1* and *Scgb3a1*. Clusters 0, 1, 5, and 7 represent club-epithelial lineages subsets. Gene expression plots demonstrating expression of IL-33 in a proportion of club cells (Cluster 1). High expression of *Foxj1* in Cluster 2 identifies club-derived ciliated cell. Other clusters (3, 4, and 6) are club-derived mesenchymal lineages. Scale bar to the right denotes normalized gene expression level of marker's gene for each cell. **d** Representative sections from histological staining of CCSP (red), IL-33 (white), β-tubulin (green), and nuclear content (DAPI, blue) in lung sections. White arrows indicate IL-33-containing club cells. **e** soluble ST2 (sST2) levels in the BAL supernatants. **f** Levels of IL-33 in homogenized lung samples. Data from *n* = 7 (**a**, **d**), 8 (**e**), and 9 (**f**) mice are representative of at least three independent series of experiments and show mean ± SEM. *$P < 0.05$, **$P < 0.01$, ***$P < 0.001$ and ****$P < 0.0001$ between NA-treated and naïve (N) WT mice using one-way ANOVA, Bonferroni post-test. Scale bar in **d** = 50 µm. t-SNE (t-distribution stochastic neighbor embedding) plots in graph **c** show data from one experiment (*n* = 1).

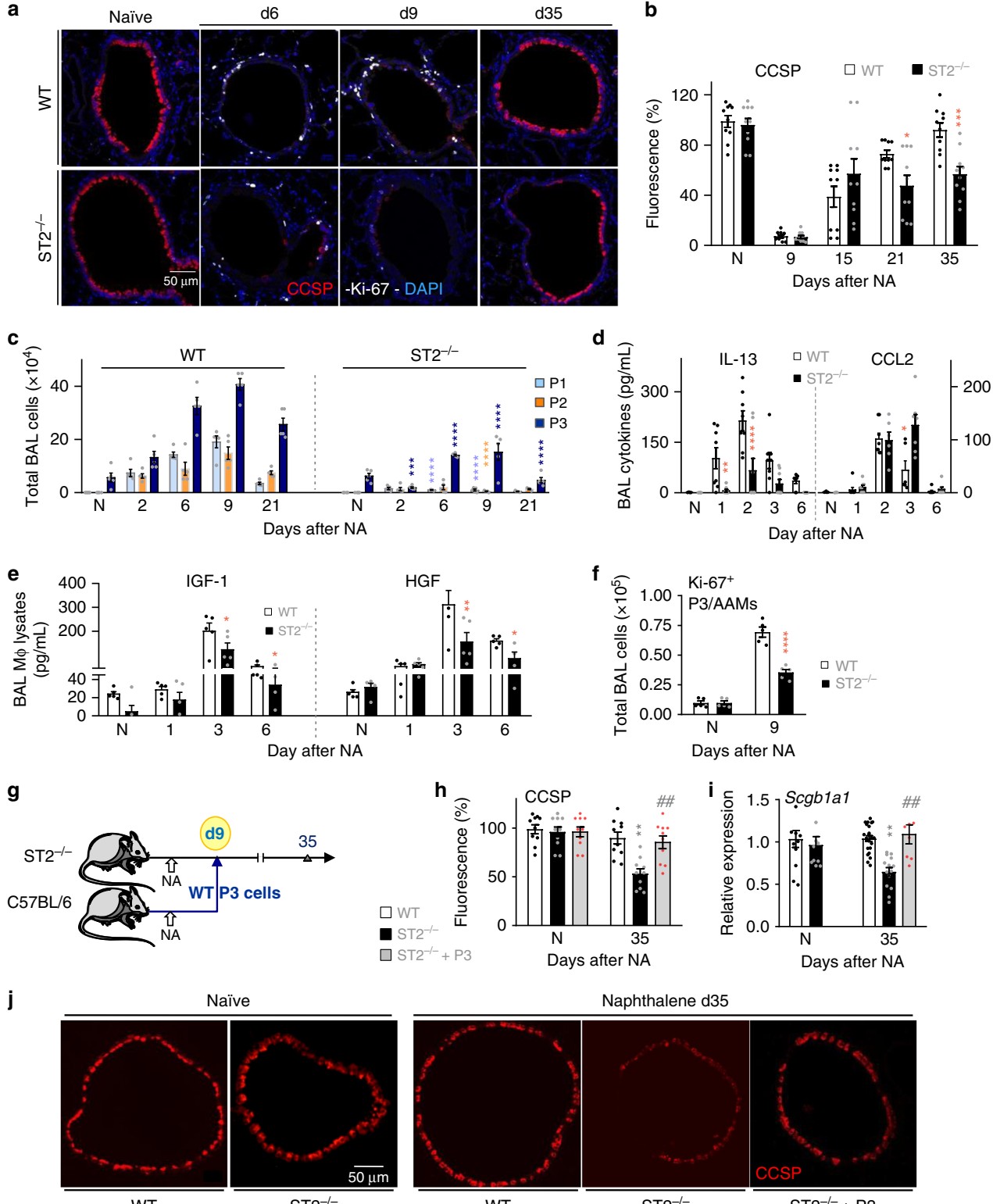

covered 25% of bronchial epithelium (d6–9; Supplementary Fig. 4b). After differentiation, mature club cells (CCSP^high) re-epithelized approximately 45% of the bronchial epithelium (d35, Supplementary Fig. 4b, left panel). Intriguingly, absence of ST2 significantly downregulated club cell proliferation and differentiation after injury, as evidenced by the decreased percentages of Ki-67+CCSP^low cells (10%; Supplementary Fig. 4b, d6, d9, right panel) and mature CCSP^high (20%; Supplementary

Fig 4b, d35, middle panel). Incomplete epithelial repair in ST2^−/− mice was associated with reduced accumulation of all myeloid subsets (P1–P3) within the airway compartment, despite the increased expression of the monocyte chemokines CCL2 and CXCL10 (Fig. 5c, d, Supplementary Fig. 4c, d).

Consistent with decreased numbers of AAMs, we found significantly lower BAL levels of IL-13 and of the AAM-associated mediators CCL17, breast-regression protein-39 (BRP-

**Fig. 5 IL-33-ST2-axis contributes to AAM-mediated bronchial reepithelialization. a** Co-immunofluorescence staining of CCSP and Ki-67 in lung tissue sections from naïve (N) and NA-injected wild-type (WT) and ST2$^{-/-}$ mice, at indicated time-points. **b** Quantification of CCSP in lung tissue sections, expressed as percentage of fluorescence within bronchioles (150–400 μm diameter). **c** Absolute numbers of P1–P3 subsets. **d** IL-13 and CCL2 levels in BAL supernatants. **e** Quantification of IGF-1 and HGF in BAL macrophage lysates. **f** Quantification of Ki-67$^+$ AAMs in BAL samples from mice. **g–j** Schematic representation (**g**) of the procedure used for P3/AAM adoptive transfer into ST2$^{-/-}$ mice on d9 after NA administration (ST2$^{-/-}$ + AAM adoptive transfer), and immunofluorescence staining of CCSP in lung tissue sections (**j**). Quantification of CCSP in lung tissue sections (**h**), expressed as percentage of fluorescence within bronchioles (150–400 μm diameter), and *Scgb1a1* mRNA levels in lung homogenates (**i**). Data from n = 10 (**a, b**), 5 (**c**), 8 (**d**), 5 (**e**), 5 (**f**), 10 (**h, j**) and 8–23 (**i**) mice are representative of at least 2–3 independent experiments and represented as mean ± SEM. $^*P < 0.05$, $^{**}P < 0.01$, $^{***}P < 0.001$ and $^{****}P < 0.0001$ between NA-treated ST2$^{-/-}$ and NA-treated WT mice using one-way (**b, e, f, h,** and **i**) and two-way (**c, d**) ANOVA, Bonferroni post-test. $^\#P < 0.05$, $^{\#\#}P < 0.01$ and $^{\#\#\#}P < 0.001$ between AAM adoptively transferred and NA-treated ST2$^{-/-}$ mice using one-way ANOVA, Bonferroni post-test. Scale bars in **a** and **j** = 50 μm.

39), CXCL12, as well as a reduction in the epithelium-regenerating factors, IGF-1 and HGF in BAL macrophage lysates from ST2$^{-/-}$ mice (Fig. 5d, e, Supplementary Fig. 4e, f). ST2 deficiency also resulted in reduced numbers of Ki-67$^+$AAMs (Fig. 5f) and lower pro-inflammatory cytokine levels in BAL macrophage lysates post NA, factors of which are critical for macrophage polarization (Supplementary Fig. 4e). To demonstrate whether P3 AAMs could also reverse the phenotype in ST2-deficient mice, we reconstituted NA-treated ST2$^{-/-}$ mice with P3 cells isolated from NA-treated WT mice (Fig. 5g). Transfer of WT AAMs completely restored bronchiolar re-epithelialization in ST2$^{-/-}$ mice (Fig. 5h–j).

**ST2 controls self-renewal and differentiation of macrophages**. To understand which biological processes in lung macrophages were regulated by ST2, myeloid cell subsets were flow-sorted from NA-treated WT and ST2$^{-/-}$ mice on d6 and their transcriptome analyzed using RNAseq. Pairwise analysis between the P1, P2, and P3 WT vs. ST2$^{-/-}$ cell populations revealed that 396, 467, and 63 genes were differentially regulated in each cell subset respectively (Supplemental Datas 8–10). Interestingly, the ST2 pathway regulated differential gene expression in each of these three myeloid subsets, which was reflected by the low overlap between the DEG (Fig. 6a, Supplementary Fig. 5a). Although the P1 subset presented a considerable number of DEG (396), pathway enrichment analysis revealed a significant alteration in cytokine-mediated signal transduction processes, such as MIF, TNF, and IL-8 signaling (Supplementary Fig. 5a), whereas, pathway enrichment analysis of the DEG within the P2 and P3 populations identified 71 and 19 ST2-dependent pathways, respectively. Those pathways downregulated in P2 ST2$^{-/-}$ cells post-NA injury were predominantly involved in cell self-renewal and differentiation processes, such as DNA synthesis/replication/repair, transcription/translation, metabolism (tricarboxylic acid (TCA) cycle, RNA, and proteins) and phagocytosis (Fig. 6b, Supplemental Data 11). Further, ST2$^{-/-}$ P3 cells exhibited drastic defects in processes controlling cell cycle, cell–matrix adhesion and migration (NCAM signaling), metabolism (RNA, protein, and glycan), vesicle-mediated transport and ECM remodeling following NA-treatment (Fig. 6c, Supplemental Data 12). More specifically, P3 cells exhibited an imbalance in ECM remodeling in favor of increased expression of proteinases, such as *Ctse*, *Adam3*, and *Mmp19*, while epithelial niche components, *Spon2*, *Sdc2*, *Sdc4*, *Eln*, *Ogn*, *Serpinb2*, and *Arg1* were significantly decreased (Fig. 6d). In addition, a dysregulated profile of growth factors promoting angiogenesis was observed, (i.e., upregulation of *Thbs1-3*, *Vgf*, and *Pdgfb*), as opposed to epithelial regeneration (i.e., decreased expression of *Areg*, *Ereg*, and *Mreg*). Furthermore, ST2 deficiency induced a dysregulated immune response in P2 and P3 subsets given the decreased expression of genes encoding *Il1a*, *Il1b*, *Il6*, and *Tnfa*, whereas genes implicated in inflammasome activation, such as *Cd300lb*[46], *P2yr10*[47], and *Nlrp12*[48] were

upregulated in both subsets (Fig. 6d, Supplemental Data 13 and 14). These findings suggest that loss of ST2 may have markedly downregulated the differentiation of P2 cells into resident P3 subset. Consequently, cells that switched into P3 phenotype exhibited significant defects in their maturation and repairing function.

To corroborate the RNAseq analysis and demonstrate an intrinsic role for ST2 on monocyte/macrophage function, we generated bone marrow-derived macrophages (BMDMs) from WT- and ST2-deficient mice. Successful cell cycle progression is necessary for effective monocyte/macrophage expansion and self-renewal[49,50], yet myeloid cells lacking ST2 exhibited a dramatic downregulation of genes involved in these processes. To determine whether ST2 regulates cell cycle of BMDMs, we examined proliferation responses to IL-33 by flow cytometry. WT BMDMs significantly proliferated in response to IL-33, i.e., a higher proportion of cells progressed into S and G2 phases when compared to unstimulated cells. However, BMDMs lacking ST2 were completely arrested within the G0/G1 phase of the cell cycle and unable to differentiate further (Fig. 7a, b). Notably, IL-13 did not impact the cell cycle under these same conditions (Supplementary Fig. 5b). We next questioned whether IL-33 modulated the expression of key genes involved in these pathways. Here, responses to IL-13 were also examined, since Lechner et al.[51] reported the requirement of CCR2$^+$ monocytes and IL-4/IL-13 signaling in the regenerating lung after pneumonectomy. Strikingly, IL-33 induced the transcription of mRNAs encoding the apoptosis inhibitors, *Api5* and *Bcl-xl*, the self-renewal mediators, *c-Maf*, *Klf4* (*Kruppel-like factor 4*), *c-Myc*, *Ccnd1* (Cyclin D1), and the transcriptional effector *E2f1*, all of which are involved in S phase progression (Fig. 7c). Furthermore, the expression of *Bcl-xl* and *c-Maf*[50] was specifically upregulated by IL-33 but not affected by IL-13, which was consistent with IL-13 having no impact on cell cycle (Fig. 7c, Supplementary Fig. 5b). In contrast, antiproliferative markers, such as cyclin-dependent kinase inhibitor *p27*, the liver X receptors α (*Lxrα*), and transcription factor *Mafb*[52] were solely expressed in ST2$^{-/-}$ BMDMs. Importantly, IL-33 also specifically enhanced the expression of genes controlling macrophage fate, including the transcription factors, early growth response protein 1 (*Egr1*), *Gfi1* (growth factor independence 1), *Stat3* and *Stat6* (AAM regulators) and *Irf4* (IFN regulatory factor 4). Furthermore, IL-33 significantly increased the transcription of genes associated with AAM polarization and maturation, such as *Cd206*, *Siglecf*, *Arg1*, *Fizz1*, *Chil3*, *Hgf*, *Tlr4*, *Il1rl1*, *Il1rAcp*, and *p53*. It is worth noting that the transcripts encoding *Il1a*, *Il1b*, and *Il6* were also upregulated by IL-33, but not modulated by IL-13. However, IL-33 sustained IL-13-mediated downregulation of genes encoding the proinflammatory macrophage markers, *Irf8* (IFN regulatory factor 8) and *Stat1*, as well as the pro-inflammatory mediators *Tnfa* and *Cxcl10* (Fig. 7c, Supplementary Fig. 5c).

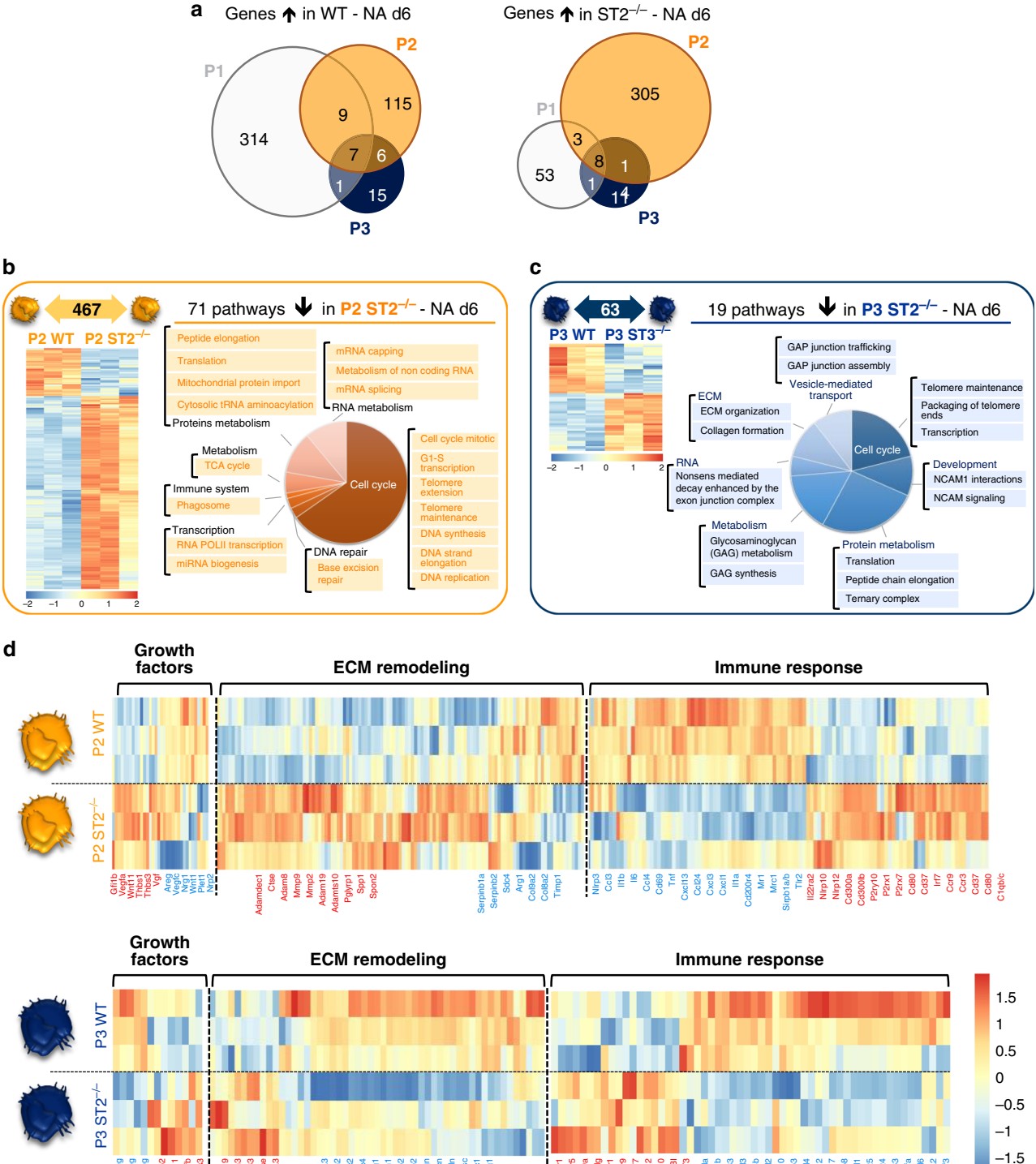

**Fig. 6 Lack of ST2 severely altered the transcriptome of airway macrophages. a–d** Transcriptomic profiling of myeloid cells flow-sorted from ST2$^{-/-}$ and WT mice 6 days after NA treatment. **a** Summary of differentially expressed genes ($P < 0.01$; FC > 2) in each pairwise comparison showing the total number of differentially expressed genes unique and shared between P1–P3. **b, c** Heatmaps and summary of the total number of differentially expressed genes ($P < 0.01$; FC > 2) in P2 (panel **b**) and P3 (panel **c**) between WT and ST2$^{-/-}$. The downregulated pathways associated with the gene differentially expressed in ST2$^{-/-}$ for P2 and P3 are illustrated on the right side of the heatmaps. **d** Heatmaps showing the top relative expression of DEG that are associated with macrophage ability to modulate growth factors secretion, extracellular matrix (ECM) remodeling and immune response regulation between P2 (left panel) and P3 (right panel). Presented in red and blue the gene upregulated and downregulated in ST2$^{-/-}$, respectively. Scale bar on the bottom denotes relative log$_2$ differences in gene expression for each row.

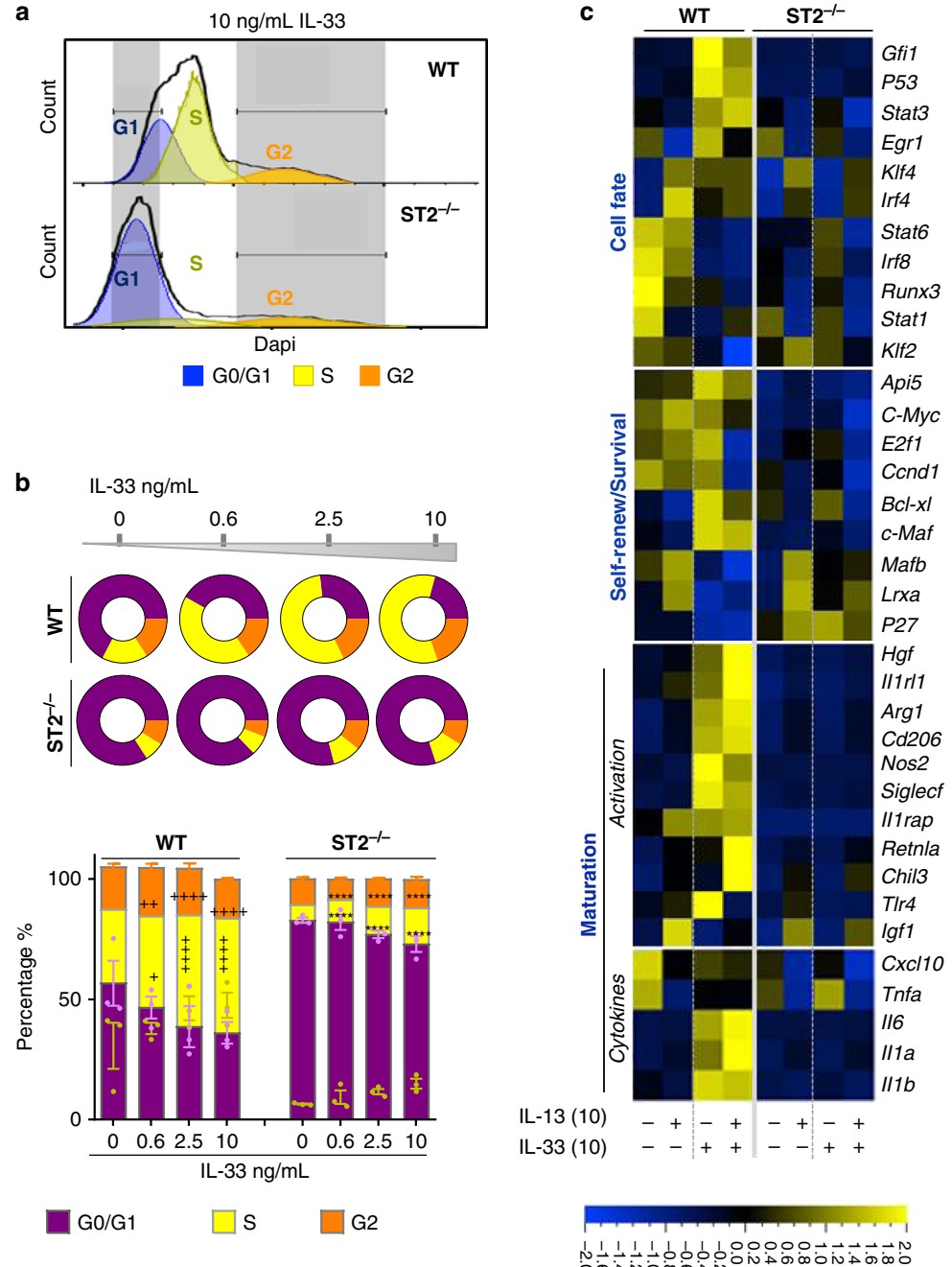

**Fig. 7 ST2 controls macrophage cell cycle progression and activation in vitro. a–c** Bone marrow-derived cells were cultured in M-CSF alone (unstimulated) or cultured in the presence of M-CSF with varying concentrations of IL-13 and IL-33 for 3 days ex vivo. **a** Representative histograms from WT and ST2$^{-/-}$ BMDMs showing DAPI staining after fixation. Cell cycle phases (G0/G1, S, and G2) are gated by nuclear DNA content. **b** Quantification of BMDMs in the three cell cycle phases and shown in (**a**) and disc graphs denote mean only. **c** Heat-map constructed from Fluidigm analysis of mRNA transcripts for the denoted genes from cultured WT and ST2$^{-/-}$ BMDMs. Scale bar on the bottom denotes relative log$_2$ differences in gene expression for each row. Samples are bone marrow treated and cultured separately from three individual mice per genotype. All bar graphs show means ± SEM. Data from $n = 3$ mice are representative from two independent experiments. $^*P < 0.05$, $^{**}P < 0.01$, $^{***}P < 0.001$ and $^{****}P < 0.0001$ between indicated IL-33-treated and unstimulated BMDMs. $^+P < 0.05$, $^{++}P < 0.01$, $^{+++}P < 0.001$ and $^{++++}P < 0.0001$ between IL-33-stimulated ST2$^{-/-}$ and WT BMDMs using two-way ANOVA, Bonferroni post-test.

Lastly, the expression of AAM markers on WT and ST2$^{-/-}$ BMDMs, following IL-33 and/or combined IL-13+IL-33 stimulation was evaluated. Both IL-33 and IL-13 induced significant ST2 (GFP) expression on CD206$^+$ BMDMs, albeit the effect of IL-33 stimulation was much greater than that of IL-13; when these cytokines were added in combination, ST2 expression was additively enhanced compared to IL-33, or IL-13 alone (Fig. 8a,

b). Although stimulated with IL-13, ST2$^{-/-}$ BMDMs failed to acquire an AAM phenotype as evidenced by the diminished frequency of CD206$^+$Arg-1$^+$ cells (Fig. 8c, d). Similarly, lack of ST2 not only downregulated HGF secretion in IL-33-induced BMDMs but also markedly reduced HGF production upon IL-13 stimulation (Fig. 8e). In this setting, IGF-1 production was not affected by ST2 deficiency, whereas IL-13-induced BRP-39 release

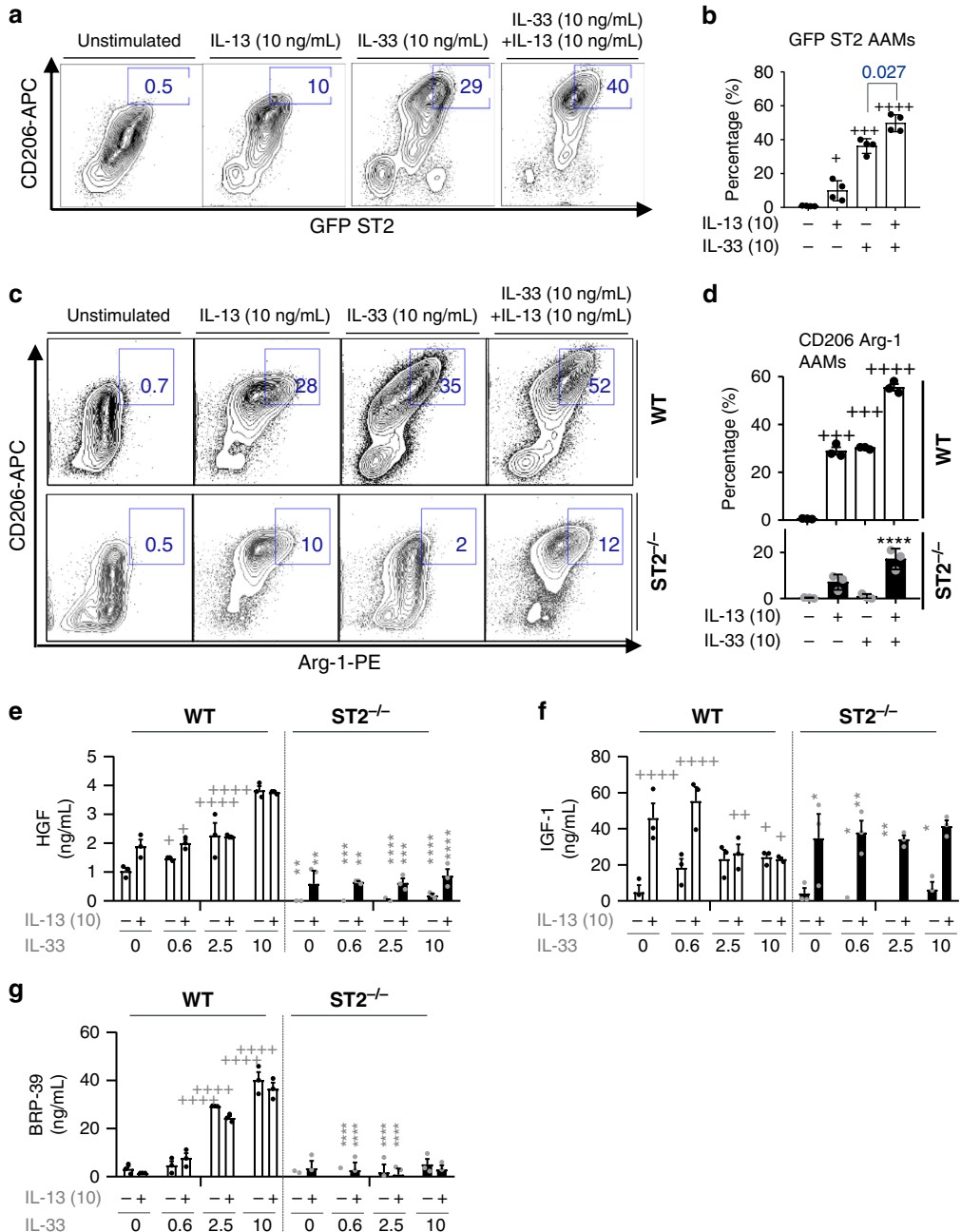

**Fig. 8 IL-33 synergizes with IL-13 to ensure a matured macrophage repairing phenotype. a–g** Bone marrow-derived cells from WT Balb/c, ST2$^{-/-}$ (Balb/c background), or ST2-GFP (Balb/c) mice were cultured in M-CSF alone (unstimulated), or cultured in the presence of M-CSF with varying concentrations of IL-13 and/or IL-33 for 6 days ex vivo. **a** Representative flow cytometric plots of CD206 and GFP expression from ST2 GFP bone marrow-derived macrophages (BMDMs). Numbers next to gate denote percentage of CD206$^+$ ST2 GFP$^+$ BMDMs. **b** Quantification of the frequency of CD206$^+$ GFP ST2$^+$ gated in (**a**). **c** Representative flow plots of Arginase-1 (Arg-1) and CD206 expression on WT and ST2$^{-/-}$ BMDMs. Gates and numbers denote the percentage of CD206$^+$ Arg-1$^+$ BMDMs. **d** Quantification of the frequency of CD206$^+$ Arg-1$^+$ BMDMs from WT or ST2$^{-/-}$ mice as gated in (**c**). **e–g** Quantification of HGF (**e**), IGF-I (**f**), and BRP-39 (**g**) from the supernatants of WT and ST2$^{-/-}$ BMDMs. Samples are bone marrow treated and cultured separately from three individual mice per genotype. Bar graphs from $n = 4$ (**b**), 3 (**d–g**) mice, show mean ± SEM pooled from three independent experiments.

was entirely dependent on ST2 (Fig. 8f, g). IL-33 has been previously reported to amplify AAM polarization and chemokine production[19], however, our data confirm and extend an intrinsic role for ST2 on macrophage function, as well as highlight an important synergy between IL-33 and IL-13, whereby they can both impact macrophage function in an additive or synergistic manner depending on the mediator examined (Fig. 7c, Fig. 8a–d). In short, we demonstrate an upstream role of IL-33-ST2 in macrophage differentiation and maturation where synergy with IL-13 may potentiate macrophage activation and their ability to produce growth factors associated with epithelial repair.

**IL-33 promotes ILC2 activation for macrophage-induced repair.** ILC2s have also been implicated in epithelial repair following influenza-induced injury[41,53–55] and mediate lung

immunity at barrier sites by actively responding to IL-33[15,53,54,56,57]. Recently, Lechner et al.[51] demonstrated that both ILC2-derived IL-13 and IL-4/IL-13 signaling within the hematopoietic compartment are required for optimal lung regeneration after pneumonectomy. Similarly, we hypothesized that IL-33 was upstream of IL-13 and that an ST2-ILC2 axis may orchestrate bronchial repair by providing IL-13 and regulating AAM polarization post NA injury. Thus, we performed the following experiments to validate our hypothesis. ILC2s were easily identifiable in the lungs of naïve and NA-treated mice as cells lacking CD3 and CD49b, as well as other lineage markers, except for CD90[55] (Supplementary Fig. 6a). Indeed, we observed a small, but significant increase in the frequency of lung ILC2s post NA but this did not reflect an appreciable increase in total ILC numbers (Fig. 9a, Supplementary Fig. 6a). Unlike other forms of lung injury, such as cigarette smoke exposure and viral infection[15,55], ILC2s from NA-treated mice maintained their type 2 phenotype, producing significantly more IL-13, GM-CSF, and IL-5, when compared to naïve mice (Fig. 9a–c, Supplementary Fig. 6b–d). We examined all cells over the course of injury, specifically IL-13 producing cells, that were present in the lung. In this setting, IL-13$^+$ lung NK or total T cells, CD4$^+$ and CD8$^+$, numbers were unaltered (Supplementary Fig. 6e–h), supporting our hypothesis that ILC2s are the main source for IL-13 production post injury. Lung regulatory T cells (Treg cells) and ILC2s have been shown to mediate lung tissue repair following influenza-induced damage, the latter cells through the production of amphiregulin[41,58]. Post injury, we found no significant differences in the number of lung-associated Treg cells (CD4$^+$ CD25$^+$ CD44$^+$ cells) when compared to those from naïve mice (Supplementary Fig. 6i, j). Furthermore, we quantified the levels of IL-13 and amphiregulin in the supernatants of stimulated ILC2s, isolated from the lungs of NA-treated mice; while the amounts of IL-13 were markedly upregulated, we were unable to detect significant changes in amphiregulin levels at several timepoints post NA (Fig. 9c). Together, these data suggest that ILC2s were activated early following epithelial damage and responded by producing type 2 cytokines. To demonstrate that ILC2s contribute to AAM activation post-injury, lung GFP$^+$ILC2s were adoptively transferred into $Rag2^{-/-}/Il2r\gamma c^{-/-}$ mice, which lack T, B, NK cells and ILCs, 2 days prior to NA (Fig. 9d). Successful transfer was determined by the presence of GFP$^+$ILC2s in recipient mice (Supplementary Fig. 7a, b). When compared to WT NA-treated animals, $Rag2^{-/-}/Il2r\gamma c^{-/-}$ mice exhibited an altered epithelial repair, i.e., a decrease in lung expression of Scgb1a1 mRNA (Fig. 9e). Concomitantly, we observed a significant decrease in total BAL cell numbers (Fig. 9f, g), which associated with an altered distribution of ST2$^+$ macrophage populations within the lung. This was accompanied by an increase in P1 cell numbers and a significant reduction in ST2$^+$ P2 and ST2$^+$ resident P3 macrophages (Fig. 9f–h, Supplementary Fig. 7c). Furthermore, mice that lacked ILC2s had dramatically decreased lung levels of IL-13 (Fig. 9i, j) and lower amounts of the AAM-associated markers, BRP-39 and CCL17, and epithelium growth factors, HGF and IGF-1 (Fig. 9k, Supplementary Fig. 7d). Reconstitution of ILC2s to the lungs of $Rag2^{-/-}/Il2r\gamma c^{-/-}$ mice restored club cell regeneration and myeloid cell populations, including the number of ST2$^+$ resident macrophages, to levels observed in C57BL/6 NA-treated mice, similarly IL-13 levels and growth factors associated with AAM and repair were also restored (Fig. 9e–k, Supplementary Fig. 7c, d). Lastly, we showed that BMDMs derived from $Rag2^{-/-}/Il2r\gamma c^{-/-}$ mice were able to differentiate into Arg-1$^+$ CD206$^+$ cells expressing ST2 (Supplementary Fig. 7e, f) and importantly produce significant levels of CCL17, IGF-1, and HGF in response to combined IL-33 and IL-13 stimulation (Supplementary Fig. 7g). Thus macrophages are functional in

$Rag2^{-/-}/Il2r\gamma c^{-/-}$, (ILC-deficient), mice and cytokine production in ILC2 recipient $Rag2^{-/-}/Il2r\gamma c^{-/-}$ animals was not due to contamination of B, T, or NK cells. Collectively, these findings highlight the importance of ILC2-derived IL-13 and the synergistic role which this cytokine plays with IL-33 in myeloid cell differentiation and effective macrophage activation into the AAM phenotype, both of which are essential for epithelial repair. Notably, the IL-33 pathway appears to be upstream from ILC2s, being a potent activator of myeloid cells and required for maintaining IL-13 production post NA injury.

## Discussion

This study unveils a protective role for the IL-33-ST2 pathway in club cell regeneration. By combining depletion and reconstitution experiments with transcriptomic analyses, we provide evidence that this axis is central to bronchial re-epithelialization due to the intrinsic role of ST2 in myeloid cell differentiation, self-renewal and maturation of competent repairing macrophages. Additionally, we show this axis also regulates and maintains ILC2 function, which is required for inducing AAM polarization and epithelial growth factor secretion.

Our study revealed three distinct populations of myeloid cells accumulating within sites of injury. Here we identified inflammatory monocytes (P1) that infiltrated the lung and differentiated into short-living recruited cells, (P2), which upon the appropriate signals, readily switched into resident AAMs (P3). As shown previously[9,49] and here, both infiltrating and resident macrophages exhibited enhanced proliferation maintaining optimal cell numbers within the lung[12] during maximal club cell proliferation. The macrophage depletion and reconstitution experiments highlight a significant and specific role for P3 cells in bronchial re-epithelialization and, our transcriptional profiling identified several mechanisms by which AAMs may promote club cell regeneration. These include the secretion of growth factors and ECM components, which have been shown to promote epithelial regeneration[31–36,59]. By contrast, P2 cells were phenotypically more profibrogenic and proinflammatory, resembling macrophages observed in fibrosis[26]. More importantly, GO pathway analysis between P2 and P3 subsets, identified upregulation of cell cycle, immune-modulation and efferocytosis processes in AAMs, whereas pathways enriched in P2 cells were mostly associated with antigen presentation, complement activation, and signal transduction.

Of significance, the IL-33/ST2 pathway was upregulated within cluster I (P1 and P2 subsets) upon macrophage differentiation/ maturation. Predominantly considered a pathogenic pathway contributing to type 2 inflammation, this axis can also amplify type 1- and type 17-mediated pathology[40,60,61]. Here, we describe another, albeit paradoxical, role for this complex axis in regulating bronchial epithelial repair. Recent reports have linked IL-33-ST2-axis to AAM-mediated tissue repair in other organs[42,62,63], our study extends these observations and provides evidence for a nonredundant role for IL-33/ST2 in airway macrophage maturation and differentiation to support the regenerative niche. Notably, ST2 deficiency leads to an incomplete re-epithelialization post NA that was associated with reduced AAM activation and proliferation, despite the presence of significant amounts of CCL2 and CXCL10[64]. Consistent with decreased AAM numbers, ST2 deficiency results in a loss of lung IL-13 levels, which was accompanied by a significant accumulation of P1 cells. Furthermore, in support of their defective reparative ability, macrophages lacking ST2 produced significantly lower levels of the epithelial growth factors, IGF-1, HGF, and of CXCL12 which were associated with a decrease in the proportion of club cell progenitors required for re-epithelialization. In

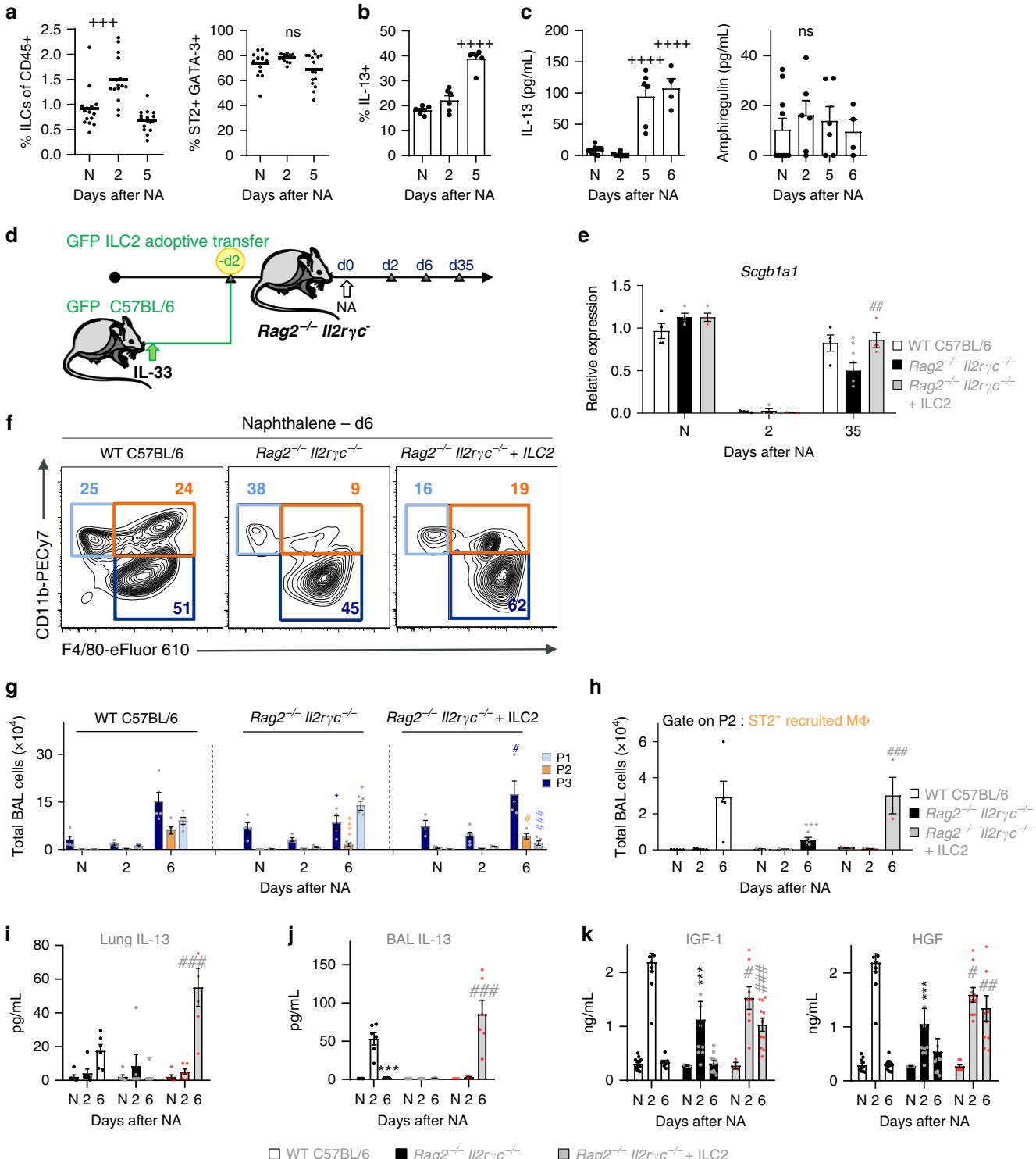

parallel, our transcriptomic analysis of ST2-deficient myeloid subsets post injury revealed drastic alterations of the cell cycle, cell–matrix adhesion and migration, metabolism and ECM remodeling processes in P2 cells. These alterations likely prevented recruited P2 cells from acquiring a functional reparative P3 phenotype to restore the damaged epithelium. More specifically, ST2$^{-/-}$ P3 cells exhibited impaired metabolic responses and dysregulated profiles of ECM components, in addition to growth factors essential for the regenerative niche. The complete bronchial re-epithelialization following reconstitution of P3 cells to injured ST2-deficient mice, points to a potential intrinsic role for ST2 in regulating macrophage maturation and reparative function.

Indeed, our studies with mouse BMDMs uncovered a crucial role for ST2 in myeloid cell differentiation, self-renewal and survival, which is consistent with a report demonstrating that ST2 is required for hypertrophic growth of chondrocytes[65]. BMDMs from ST2-deficient mice exhibit a significant arrest within the G0/G1 phase of their cell cycle, failing to differentiate further, this was associated with reduced expression of transcripts encoding for anti-apoptotic and self-renewal mediators. Interestingly, IL-13 failed to alter macrophage division, aligning with a recent study

**Fig. 9 Lung ILC2s produce IL-13 after NA injury and contribute to macrophage maturation. a–c** Quantification of the frequency of ST2$^+$ GATA-3$^+$ expressing ILCs (**a**) found in the lung at days 0, 2, and 5 after NA-treatment; N = naïve. ILCs were isolated from the lungs of mice on days 0, 2, 5, and 6 post-NA-treatment and stimulated ex vivo, then flow-stained for IL-13 production and quantified as frequency (**b**). IL-13 and amphiregulin production were quantified also in stimulated ILC supernatants (**c**). **d** Schematic for the transfer of lung ILC2s into $Rag2^{-/-}/Il2r\gamma c^{-/-}$ recipient mice followed by naphthalene administration ($Rag2^{-/-}/Il2r\gamma c^{-/-}$ + ILC2). **e** Levels of the $Scgb1a1$ mRNA in total lung homogenates in WT C57BL/6 or $Rag2^{-/-}/Il2r\gamma c^{-/-}$ mice that adoptively received GFP$^+$ ILC2s ($Rag2^{-/-}/Il2r\gamma c^{-/-}$ + ILC2). **f** Representative flow cytometric plots of P1-P3 subsets of BAL cells from WT C57BL/6 or $Rag2^{-/-}/Il2r\gamma c^{-/-}$ mice that adoptively received GFP$^+$ ILC2s ($Rag2^{-/-}/Il2r\gamma c^{-/-}$ + ILC2). Numbers nears gates denote percentage. **g** Quantification of total P1-P3 cells in BAL using the gating strategy in (**f**). **h** Quantification of the total number of ST2-expressing recruited macrophages (ST2$^+$ P2 recruited macrophages) in the BAL. **i** Levels of IL-13 in lung homogenates. **j, k** Levels of IL-13 (**j**) and IGF-1 and HGF in BAL supernatants (**k**). Data from n = 15 (**a**), 6 (**b**), 3 to 9 (**c**), 4 (**e**), 5 (**f–h**), 6 (**i**, **j**), and 8 (**k**) mice, show mean ± SEM pooled from three independent experiments. $^{***}P < 0.001$ and $^{****}P < 0.0001$ between NA-treated and naïve (N) WT mice using one-way ANOVA, Bonferroni post-test. $^*P < 0.05$, $^{**}P < 0.01$, $^{***}P < 0.001$ and $^{****}P < 0.0001$ between NA-treated $Rag2^{-/-}/Il2r\gamma c^{-/-}$ and WT mice using one-way ANOVA, Bonferroni post-test (**a–c**). $^{#}P < 0.05$, $^{##}P < 0.01$ and $^{###}P < 0.001$ between AAM adoptively transferred and NA-treated ST2$^{-/-}$ mice using one-way (**g–k**) and two-way ANOVA (**e**), Bonferroni post-test.

that ST2 promotes macrophage proliferation independently of IL-4R$\alpha$[66,67].

While both IL-13 and IL-33 regulate AAM activation[19] and display some overlapping functions, we found that transcripts encoding the biologic processes of self-renewal and macrophage differentiation/maturation, were all severely altered in BMDMs lacking ST2. In addition, upregulation of anti-proliferative markers and concomitant downregulation of apoptosis inhibitors, indicated poorly differentiated and potentially dying cells. Interestingly, IL-13 and IL-33 appear to act in tandem to promote differential aspects of repair and a balance of their local levels may be required to subtly tune the immune response for optimal epithelial regeneration.

A regulatory role for ILC2s in epithelial repair has been reported[15,53,54,56,57], here we demonstrate that the IL-33 pathway is upstream from ILC2s being a potent activator of these cells and required for maintaining their IL-13 production post NA. Previous studies have shown that noxious or infectious agents[55] induce a significant loss of type 2 markers, such as GATA3 and ST2 in ILC2s. However, following NA-induced IL-33 release, these cells maintain their phenotypic type 2 signature by producing significant amounts of IL-13 and GM-CSF. ILC2 production of these cytokines does align with a role for AAM in repair as GM-CSF is required for the maintenance and development of myeloid cells and AAMs[68], while IL-13 appears to regulate essential AAM polarization as demonstrated for pneumonectomy-induced lung regeneration[51,69]. Mice lacking ILCs ($Rag2^{-/-}/Il2r\gamma c^{-/-}$) failed to recruit optimal numbers of ST2$^+$ P2 and P3 cells. Here we observed an increased infiltration of monocytes (P1) that were unable to promote repair. $Rag2^{-/-}/Il2r\gamma c^{-/-}$ mice also had severely impaired IL-13 production and, since NK and T cells failed to produce this cytokine post-NA, ILC2s were the major lung cell source of IL-13 in this model. Consistent with the incomplete epithelial repair observed in $ST2^{-/-}$ mice, $Rag2/Il2r\gamma c$ deficiency led to a dramatic decrease in AAMs and levels of the growth factors HGF and IGF-1. While ILC2 reconstitution to $Rag2^{-/-}/Il2r\gamma c^{-/-}$ mice completely rescued this phenotype and confirmed that ILC2s play a crucial role in AAM-mediated epithelial repair. ILC2s have been directly implicated in influenza-induced alveolar repair due to their production of amphiregulin and Arginase-1[40,41,70]. Here, we were unable to detect significant changes in ILC2-derived amphiregulin post NA, although, we did detect mRNA upregulation of $Areg$ and $Arg1$ (also protein expression) in polarized AAMs. Further studies are required to determine the direct contribution of ILC2s to bronchial re-epithelialization. Collectively, these data support a critical role for ILC2s, in functioning as part of a larger, multicellular circuit that activates and differentiates monocytes into growth factor-secreting AAMs. Thus, ILC2s can mediate both inflammation and repair at barrier sites by coordinating the

activation state of specific effector cells. The synergistic effects of IL-33+IL-13 on ST2 expression and (BMDM) AAM polarization supports our hypothesis for an IL-33-ILC2-AAM-axis orchestrating repair and highlights an essential contribution of the IL-33-ST2 pathway as a potent activator of myeloid cells and requirement for maintaining IL-13 production post NA injury.

In conclusion, this report reveals a central role for IL-33-ST2 in epithelial regeneration and uncovers a function for ST2 in promoting macrophage differentiation, survival and self-renewal. Furthermore, this study highlights the complexity of the differential functions between IL-33 and IL-13 which exhibit overlapping, synergistic, and nonredundant functions in balancing the immune response towards optimal epithelial regeneration. Translating these findings to human disease will help with the design of future intervention strategies for chronic inflammatory diseases.

## Methods

**Mice.** Adult (8–13 week old) C57Bl/6J mice were supplied by Janvier Laboratory (Saint-Berthevin, France) and by Charles River (Lyon, France), respectively. ST2-deficient mice[38] were licensed from MRC for their use at MedImmune and subsequently backcrossed onto C57BL/6 background using speed congenic breeding at CRL, whereas GFP ST2 reporter mice were generated in house. Animals were housed under specific pathogen-free conditions and kept in a room with controlled temperature (~23 °C) and humidity under 12 h light/dark cycle, according to protocols approved by the Paris Nord Ethic Committee for Animal Care and Ministry of Education and Research (n°02013.01). C57BL/6J (Jax, #000664), $Rag2^{-/-}/Il2r\gamma c^{-/-}$ ($Rag2^{tm1Fwa}$ $Il2r\gamma^{tm1Wjl}$) deficient (Taconic, #4111), green fluorescent protein (GFP) transgenic (C57BL/6-Tg(CAG-EGFP)131Osb/LeySopJ, Jax, #006567) mice, and ST2 GFP (C57BL/6-Il1rl1$^{tm3548.1(T2a-EGFP)Arte/+}$)[55,71] mice were housed at Med-Immune/AstraZeneca and treated according to protocols approved by the Institutional Animal Care and Use Committee at MedImmune/AstraZeneca. For all mouse experiments, female mice were randomized and placed into groups based on weight.

**NA exposure.** NA (Sigma-Aldrich, Paris, France) was dissolved in corn oil and injected intra-peritoneally to C57Bl/6J, $Rag2^{-/-}/Il2r\gamma c^{-/-}$, and ST2$^{-/-}$ female mice at a dose of 200–250 mg kg$^{-1}$. After injection, mice were maintained in specific pathogen-free conditions, and monitored daily for signs of distress and were given gel packs to aid in the recovery.

**Macrophage depletion.** Sterile suspensions of liposomes containing either PBS (control liposomes, Encapsome®), or CL (dichloromethylene diphosphonate, 5 mg mL$^{-1}$, Clodrosome®) were purchased from Encapsula NanoSciences (Brentwood, TN, USA). Once anesthetized in a chamber containing 2.5% isoflurane in O$_2$, mice received 3 intranasal doses of liposomal CL, at 20 mg kg$^{-1}$, on d2, d5, and d8 post NA injection.

**Adoptive transfer of resident airway macrophages.** BAL was performed in NA-injected mice on d9 and AAMs were isolated by negative selection, using CD11b$^-$ beads. These AAMs ($6 \times 10^5$ per mouse) were immediately injected intra-tracheally into NA-treated, macrophage-depleted, or ST2$^{-/-}$ mice.

**Transfer of lung ILC2s to $Rag2^{-/-}/Il2r\gamma c^{-/-}$ mice.** GFP transgenic mice were treated intranasally with recombinant murine IL-33 (generated in-house) suspended in PBS at 1 μg per dose on days 0, 2, and 4. On day 5 or 6, GFP transgenic

mice were euthanized as described below and the lungs were digested. Briefly, lungs were perfused with PBS, cut into ~2 mm pieces and incubated with Liberase (42.4 µg mL$^{-1}$) and DNAseI (10 U mL$^{-1}$; both from Roche) for 45–60 min at 37 °C before being mashed through a 70 µm cell strainer and washed with complete RPMI. Remaining blood cells were lysed with ACK cell lysing buffer (Invitrogen) and single cell suspensions were obtained. Following digestion, lungs were pooled and cells were counted and biotinylated antibodies against the lineage markers CD3ε (clone 145-2C11), CD19 (1D3), B220 (RA3-6B2), CD5 (53-7.3), TCRβ (H57-597), TCRγδ (GL3), CD11c (N418), F4/80 (BM8), Gr-1 (RB6-8C5), Ter119 (TER-119), CD49b (DX5; eBioscience), NK1.1 (PK136) and CD27 (LG3A10, BioLegend), all at 25 µg mL$^{-1}$ per $2 \times 10^8$ cells, (Supplementary Table 1). Cells were then incubated with anti-biotin microbeads (Milltenyi; 1:5 dilution at $10^8$ cells per mL) and depleted following manufacturer's protocol. After magnetic sorting, the enriched ILC2s were then sorted on an Aria II Fusion (BD), washed and suspended in PBS and $10^5$ purified ILC2s were tail-vein injected into recipient $Rag2^{-/-}/Il2rγc^{-/-}$ mice in 100 µL PBS. Recipient mice were then allowed to rest for 2 days after transfer before further experiments were performed.

**Lung tissue preparation.** Mice were anesthetized by a subcutaneous injection of 0.25 mL of (v:v) ketamine 50 mg mL$^{-1}$ and xylazine 20 mg mL$^{-1}$ (Centravet, Nancy, France) and exsanguinated via the inferior vena cava. The right upper and lower lung lobes were resected and homogenized in 1 mL Tris/EDTA/Tween buffer containing protease inhibitors and TriZol (Life Technologies, Saint Aubin, France) for protein and total RNA isolation, respectively. The right median and accessory lobes were cryopreserved in liquid nitrogen and stored at −80 °C. The left lung lobe was inflated and fixed in 4% buffered formaldehyde for 24 h at 4 °C and processed for obtaining paraffin-embedded blocks. Five micrometre lung tissue sections were either stained with hematoxylin and eosin for morphological examination, or they were used for immunohistochemistry.

For experiments involving flow cytometry, lungs were perfused with PBS, cut into ~2 mm pieces and incubated with Liberase (42.4 µg mL$^{-1}$) and DNAseI (10 U mL$^{-1}$; both from Roche) for 45–60 min at 37 °C before being mashed through a 70 µm cell strainer and washed with complete RPMI. Remaining blood cells were lysed with ACK cell lysing buffer (Invitrogen) and single cell suspensions were obtained for further analysis.

**BAL cell preparation.** Mice were terminally anesthetized by a subcutaneous injection of ketamine and xylazine, as described above. BAL fluid was obtained by injecting and recovering $2 \times 1$ mL of sterile PBS through a tracheal catheter. After centrifugation (400$g$, 5 min, 4 °C), aliquots of cell-free supernatant were stored at −80 °C until mediator analysis (see below). Cell pellets were resuspended, either in PBS for FACS analysis, or in TriZol, for total RNA isolation. Total BAL cells and macrophage counts were determined on cytospin preparations after *Diff-Quik* staining (Polysciences Europe, Eppelheim, Germany), by counting at least 200 cells in randomly-selected fields. Images were captured with a 63× oil immersion objective.

**Immunohistochemistry.** Serial 5 µm-thick paraffin sections were deparaffinized in xylene, followed by rehydration in ethanol washes. Antigen retrieval was performed by boiling for 15 min in 10 mM sodium citrate buffer, at pH 6. Endogenous peroxidase activity was inhibited using peroxidase blocking solution (Dako, Trappes, France) and slides were incubated overnight at 4 °C with F4/80 antibody (Supplementary Table 1). Antibody-antigen complexes were detected using biotinylated secondary antibodies, followed by avidin–biotin–horseradish peroxidase or avidin–biotin–alkaline phosphatase complex (ABC), and diamino-benzidine or alkaline phosphatase substrates (all from Vector Labs, Eurobio/Abcys, Les Ulis, France). Sections were counterstained with nuclear hematoxylin. Images were captured at 20× objective.

**Immunofluorescence.** Immunofluorescence for F4/80/Ki-67, CCSP/Ki-67, and IL-33/CCSP/β-tubulin were performed on paraffin-embedded lung tissue sections after deparaffinization in xylene and rehydration in ethanol washes. Tissue auto-fluorescence was minimized using 0.1% Sudan Black (Sigma-Aldrich, Paris, France). Sections were incubated with primary antibodies, (Supplementary Table 1). IL-33 staining was amplified using Tyramide Signal Amplification (TSA™) from PerkinElmer, according to the manufacturer's protocol. For co-localization experiments, secondary donkey antibodies conjugated with Alexa Fluor 488, Alexa 568 or Alexa 647 (Invitrogen/Molecular probes, Saint Aubin, France) were used. Sections were counterstained with nuclear Dapi. Fluorescent images were obtained using a confocal Leica TCS SP8 microscope and LAS AF (Leica Application Suite Advanced Fluorescence, Paris, France) microscope software. Images were captured with 25× and 40× oil immersion objectives.

**CCSP quantification.** Bronchioles (diameter = 150–350 µm) were examined for CCSP expression in Naïve, NA-treated, depleted and transferred mice. Mean intensity of fluorescence per bronchiole was used as quantitative dataset and normalized to the mean fluorescence intensity of CCSP expression in naive mice to calculate the total CCSP expression per bronchiole. Then, the mean intensity of fluorescence was quantified within each club cell. Club cells displaying less or

higher than 50% of the mean fluorescence intensity of the reference, (CCSP expression in club cells of naïve mice), were qualified as CCSP$^{low}$ or CCSP$^{high}$, respectively. Finally, Ki-67$^+$ cells were counted within CCSP$^{low}$ cells.

**Flow cytometry analysis of BAL monocytes and macrophages.** BAL and lung cells from naïve, NA-injected mice, administered with PBS- or CL-containing liposomes, were fixed and permeabilized using cytofix/cytoperm buffer (R&D systems, Lille, France). Flow cytometry (Canto II and LSRII/Fortessa, Becton Dickinson, Paris, France) was performed on fixed $0.5 \times 10^6$ BAL cells, after incubation with 50 µL of Fc block anti-mouse CD16/CD32 (Becton Dickinson Pharmingen) on ice for 15 min, followed by staining with anti-mouse antibodies, (Supplementary Table 2), according to manufacturer's protocols, as well as their isotype controls. Biotinylated antibody-antigen complexes were detected using streptavidin conjugated with Brilliant violet 421 (Biolegend, Yvelines, France). Cell cycle was examined using Propidium Iodide (Biolegend). FACS data was acquired using BD LSR Fortessa™ Flow Cytometer and were analyzed by Flow Jo 9.5 software. After gating out debris and neutrophils (F4/80$^{low}$CD11b$^{hgh}$CD64$^{low}$), monocytes and macrophages were analyzed on CD64 gate. Resident and recruited macrophages were identified as FSC$^{high}$F4/80$^{high}$CD11c$^{high}$CD11b$^−$ and FSC$^{int}$ F4/80$^{int}$ CD11b$^+$ Ly6C$^{low}$ CCR2$^{high}$ or ST2$^{high}$, respectively. Inflammatory monocytes were defined as F480$^{low}$CD11b$^{high}$Ly6C$^{high}$CD11c$^−$. Apoptotic macrophages stained positive for annexin V (Biolegend, Yvelines, France). Viable cells were counted before staining using the trypan blue exclusion method. The absolute numbers of different cell types were calculated based on the proportion of events analyzed by FACS and related to the total number of viable cells enumerated in each sample. Fixable viable dye (Affimetrix, eBioscience) was added before fixing cells to determine the total number of dead cells.

**Identification of ILCs.** Lung homogenates, MACS-treated ILCs, and ex vivo stimulated ILCs underwent staining with CD3ε (clone 145-2C11), CD49b (DX5), CD45 (30-F11), CD25 (PC61), CD90.2 (53-2.1), CD44 (IM7), IL-18Rα (P3TUNYA), ST2 (DJ8). The lineage cocktail included antibodies against TCRβ (H57-597), TCRγδ (GL3), CD5 (53-7.3), F4/80 (BM8), CD11c (N418), Gr-1 (RB6-8C5), CD19 (1D3), FCε RI (MAR-1), B220 (RA3-6B2), (eBiosciences), NK1.1 (PK136), and CD27 (LG3A10), Supplementary Table 2. For intracellular transcription factor staining, cells were fixed and permeabilized (FoxP3 stain/transcription factor staining buffer set, eBioscience) according to manufacturer's instructions and stained with GATA-3 (TWAJ, eBioscience).

**Ex vivo stimulation of ILCs.** ILCs were isolated from murine lung by magnetic enrichment detailed in the ILC transfer section above and stimulated overnight with IL-2 (Peprotech) and IL-33 (generated in-house) at 25 ng mL$^{-1}$ each. Supernatants were collected and cells were restimulated with PMA, ionomycin, and Brefeldin A for 4 h. Cells were washed, stained for extracellular markers, fixed and permeabilized (BD Cytofix/Cytoperm) and stained for IL-13 (clone 13A; 1:200). Cells were analyzed by flow cytometry and supernatants were assayed for cytokine production.

**BMDMs culture and stimulation.** Femurs, tibiae, fibulae, and ilia were isolated from C57BL/6J, Balb/c, ST2-GFP, or ST2$^{-/-}$ mice on the Balb/c background. The bones were cut and marrow was flushed out with sterile medium using a needle and syringe. RBC lysis was then performed using ACK buffer and cells were seeded at $4 \times 10^5$ cells per 24-well in non-tissue culture, (for flow cytometry), or tissue culture treated, (for RNA), dishes in DMEM containing 20% FBS (v/v), 1% penicillin/streptomycin (v/v), 25 ng mL$^{-1}$ M-CSF and the indicated concentrations of IL-13 and IL-33, alone or in combination. Cells were fed on day 4 by removing half of the volume of medium and adding fresh medium with the appropriate cytokines. For flow cytometry, cells were lifted in cold PBS with 2 mM EDTA, then fixed and stained for the cell cycle study or AAM activation at day 4 and day 6, respectively. Supernatants were collected from day 6 for cytokine quantification, (BRP-39, CCL-17, HGF, IL-13, and IGF-1). FACS antibodies are listed in Supplementary Table 2. DAPI (Invitrogen) was used to label DNA for the cell cycle study. For RNA isolation, medium was removed from each well and cells were lysed directly in RLT and treated with the RNA-plus kit, as indicated by manufacturer's protocol (Qiagen).

**RNA isolation, real-time PCR, and Fluidigm.** RNA was isolated from BAL cells, or from the right lower lobe of naïve and NA-injected mice, with PBS- or CL-containing liposomes. RNA was extracted using TRIzol and RNeasy mini columns, (Ambion, Life Technologies, Paris, France), according to the manufacturer's protocol, followed by quantification using a Nanodrop Spectrophotometer (Thermo Scientific, Paris, France). In a separate series of experiments, cDNA was synthesized from 1 µg or 250 ng of total RNA obtained from whole lung homogenates or BAL cells, respectively. BAL cell and lung tissue cDNA was diluted 1:15 and 1:25, respectively, in RNase-free H$_2$O. Quantitative PCR using Taqman probes (Applied BioSystems, Paris, France) was performed using the primer sets listed in Table 3. A probe for *Gapdh* mRNA was used for normalization. Taqman Real-Time PCR system was assessed on an Applied Biosystems 7500 Fast Real-Time PCR System (Applied Biosystems). Mean values from three independent experiments were

calculated. For fluidigm, qPCR was performed using the fluidigm Biomark Dynamic array loaded with the probes found in Supplementary Table 3 (Fluidigm Corp, South San Francisco, USA).

**Single-cell RNAseq of mouse bronchial epithelial cells**. Mouse bronchioles were collected, and single cell suspensions were prepared as described in lung tissue preparation above and enriched for Epcam positive cells. For single-cell RNA-sequencing, we followed the manufacturer's protocol (Chromium™ Single Cell 3′ Reagent Kit, v2 Chemistry) to obtain single cell 3′ libraries for Illumina sequencing. Libraries were assessed for quality (TapeStation 4200, Agilent), and then sequenced on NextSeq 500 or HiSeq 4000 instruments (Illumina). Initial data processing was performed using the Cell Ranger version 2.0 pipeline (10x Genomics). Post-processing, including filtering by number of genes expressed per cell, was performed using the Seurat package V2.2 and R 3.4 and visualization was generated using UCSC Cell Browser. Clustering was realized with t-Distributed Stochastic Neighbor Embedding (t-SNE). Identification of cell clusters in mouse bronchial epithelial cells was guided by marker genes. *Scgb1a1*, *Scgb3a1*, *Foxj1*, and *Il33* gene expression was performed for each cluster. t-SNE plots were generated using Seurat.

**RNA-sequencing and bioinformatics analysis**. RNA sequencing was performed using Illumina HiSeq 2000 platform and sequence data (fastq) was quality checked using FastQC and further aligned to mouse genome (mm10) using Hisat2 (version 2.0.2). Gene count table was generated using HTSeq and further normalized to count per million (CPM) using edgeR. Differential expression gene analysis was performed using edgeR and genes with false-discovery rate < 0.05 and fold change (FC) > 2 were considered as significant if not specified separately. Heatmap visualization was built using pheatmap in R after removing low expressing genes, (CPM < 150 if not specified separately). CPM in log scale was used to generate heatmaps for growth factor, extracellular modeling and immune response and Z-score was used to generate heatmaps with DEG. Gene signatures were clustered using hierarchical clustering based on Euclidean distance. Tree was cut based on dendrogram using R cutree function. Pathway enrichment analysis for gene cluster was performed by metabase R package (v4.2.0, Clarivate Analytics) and process networks constructed by Thomson Reuters. Detailed pathway was classified based on network ontology, (Supplemental Tables 4 and 5). Gene Set Enrichment Analysis (GSEA) was performed in GenePattern using curated gene sets from Reactome database (c2.cp.reactome), and detailed pathway was further classified according to Reactome ontology (reactome.org/).

**Quantification of cytokines and growth factors**. The levels of cytokines were measured by multiplex magnetic bead immunoassay system, (Milliplex MAP kit, Merck Millipore, Molsheim, France), in BAL supernatants that had been previously concentrated 5× using Amicon Ultra-centrifugal filters, (Merck Millipore). Mouse 17-plex cytokine/chemokine detection kit (G-CSF, GM-CSF, IFN-γ, IL-1α, IL-1β, IL-4, IL-6, IL-10, IL-13, CXCL10/IP-10, CXCL1/KC, CCL2/MCP-1, CCL3/MIP-1α, CCL4/MIP-1β, CCL5/RANTES, and TNF) was used for cytokine quantification in BAL supernatants of naïve and NA-injected mice, whereas Mouse 8-plex cytokine/chemokine detection kit (Milliplex MAP kit) (IL-1α, IL-1β, IL-6, IL-10, IL-13, CXCL10/IP-10, CXCL1/KC, and CCL2/MCP-1) was used for cytokine quantification in BAL supernatants of mice administered with PBS- or CL-containing liposomes. Beads were analyzed with Luminex MAGPIX, (Merck Millipore). The detection limits for these cytokines was 3.2 pg mL⁻¹.

**Chemokine production in BAL cell lysates**. Naïve and NA-treated mice were euthanized and BAL fluid was obtained by injecting and recovering 2 × 1 mL of sterile PBS through a tracheal catheter. After centrifugation, (400*g*, 5 min, 4 °C), BAL cell pellets were resuspended at 5 × 10⁵ cells per mL in RPMI medium and seeded in 24-well plate (500 µl per well) overnight to allow macrophage adhesion to the well bottom. After 24 h, adherent macrophages were lysed with RIPA buffer, centrifuged at 13000 × *g* for 10 min, and supernatants were collected and stored at −20 °C. Samples were tested for the presence of chemokines with a Proteome Profiler Mouse Chemokine Array Kit (R&D Systems), according to manufacturer instructions. Membranes were developed for 10 min and digitized for analysis on ImageJ. Densitometry was performed using particle analysis, with measurements for each sample spot averaged and then normalized to membrane reference spots.

**ILCs and monocyte cytokine/chemokine quantification**. BRP-39, CCL-17, HGF, IL-13, and IGF-1 were quantified in BAL supernatants of naïve and NA-treated mice and supernatants of cultured monocytes by ELISA (R&D), according to the manufacturer's instructions. IL-5 and GM-CSF from cultured ILCs was determined by multiplex array, (Meso Scale Delivery). IL-13 from cultured ILCs was determined by ELISA (R&D).

**IL-33 and sST2 quantification**. The right upper lung lobes of naïve and NA-injected mice were homogenized in 1 mL lysis buffer, and the supernatant was collected for IL-33 quantification, (R&D Systems), in standardized sandwich ELISA, according to the manufacturer's protocol. BAL levels of sST2 were measured by ELISA (R&D Systems).

**Statistical analysis**. Statistical analyses were performed using GraphPad Prism v8.0.1. ANOVA with Bonferroni post-test was used to determine statistical significance between the groups in experiments involving more than two groups or time-points. The data were considered significant when *p* values were ≤0.05. For all graphs and experiments, *\*P* < 0.05, *\*\*P* < 0.01, *\*\*\*P* < 0.001, *\*\*\*\*P* < 0.0001 comparing the treated group or timepoint the mark is directly above to the naïve or unstimulated for that same genotype of mouse or cells; ⁺*P* < 0.05, ⁺⁺*P* < 0.01, ⁺⁺⁺*P* < 0.001, ⁺⁺⁺⁺*P* < 0.0001 comparing WT mice or cells to deficient, or depleted mice or cells for that same timepoint or condition (e.g., comparing WT to ST2⁻/⁻ IGF-1 levels 5 h post NA treatment); #*P* < 0.05, ##*P* < 0.01, ###*P* < 0.001, ####*P* < 0.0001 comparing add-back mice, (NA + AAM adoptive transfer or Rag2⁻/⁻/Il2rγc⁻/⁻ +ILC2), to the appropriate depleted, (NA + macrophage depletion), or gene-deficient (ST2⁻/⁻, Rag2⁻/⁻/Il2rγc⁻/⁻) mice for that condition or timepoint. Supplementary Table 6 lists the statistical analyses methods used throughout this manuscript.

Where feasible, test samples were blinded to the operator, i.e., mRNA and protein analyses. Immunohistochemical scoring was also performed blinded. In some cases, blinding of test samples was not applicable, as samples were processed identically through standard and in some cases automated procedures (RNA sequencing and analyses) but these processes do not facilitate bias outcomes.

**Reporting summary**. Further information on research design is available in the Nature Research Reporting Summary linked to this article.

## Data availability

Sequence data that support the findings of this study have been deposited in the Gene Expression Omnibus with accession numbers, GSE155356 and GSE155359, for cell-sorted bulk RNAseq data for lung monocyte/macrophage subsets from NA-treated WT mice (Fig. 3), and NA-treated ST2⁻/⁻ (Fig. 6). scRNAseq data are deposited with accession numbers, GSE155261, for subtyping naïve mouse bronchial epithelial cells (Fig. 4). The data that support the findings from these studies are available from the corresponding authors upon request. Source data are provided as a Source Data file. Source data are provided with this paper.

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

## Acknowledgements

The authors wish to thank to Nassima Ferhani, Sophie Kozelko, and Sophie Moog (Inserm UMR1152) for their help with the NA experimentation, and the *Investissements d'Avenir* program ANR-11-IDEX-0005-02, Sorbonne Paris Cité, Laboratoire d'Excellence INFLAMEX. Yoichiro Ohne for critical reading of the paper.

## Author contributions

R.D. and M.P. conceived the initial ideas for this project, R.D. designed the NA-induced lung injury study and myeloid cell characterization, performed experiments, analyzed the data, and wrote the paper. A.M.C. performed ILC the experiments, ILC and myeloid cell sorting, analyzed the experiments and discussed them in the paper. V.B. contributed to the analyses and interpretation of in vivo study. M.M. set the NA-induced lung injury model in Inserm U1152 lab. R.D. and A.B. set the NA-induced lung injury model in MedImmune/AstraZeneca. F.H. performed the analyte measurement in BAL fluids from naïve and NA-treated mouse. G.G. contributed to FACS analysis. Y.S. performed the bulk RNAseq experiments. R.D., J. Wang and J. Wu analyzed the bulk RNAseq. R.D. and X.Q. performed the single-cell analyses. R.R. supported the transcriptomic studies. A.M., R.K., A.A.H., and M.P. designed the study, analyzed the data, and wrote the paper. MedImmune/AstraZeneca provided funds to M.P. for these studies.

## Competing interests

The authors R.D., A.M.C., A.B., J.W., J.W., X.Q., Y.S., R.R., R.K., and A.A.H. are/were employed by and shareholders of AstraZeneca. The author M.P. received funding from MedImmune/AstraZeneca. The remaining authors declare no competing interests.
