## [Peer Review File · Nature Communications]

Editorial Note: This manuscript has been previously reviewed at another journal that is not operating a transparent peer review scheme. This document only contains reviewer comments and rebuttal letters for versions considered at Nature Communications. Parts of this Peer Review File have been redacted as indicated to maintain the confidentiality of unpublished data.

Reviewers' comments:

Reviewer #1 (Remarks to the Author):

In this amended manuscript Dagher et al. report that ST2-deficient mice show an impaired repair, notably club cell proliferation, in the naphthalene-induced lung injury. They show, as previously (1) the pivotal role of macrophages in this repair process, and they show how important the presence of ST2 is, and (3) they provide evidence that group 2 innate lymphoid cells are involved in promoting macrophage differentiation and proliferation in an IL-33 dependent manner.

As mentioned in the previous review, this manuscript describes in depth the role of different macrophage subpopulations in orchestrating lung epithelial barrier cell repair in this particular injury model. This model is characterized by a rapid loss of barrier cells and an early onset of tissue repair rather than by a pronounced local inflammatory response.

The authors have addressed the points raised by this reviewer in detail. Some of them satisfactory, others not so satisfactory. In total, the manuscript is better to follow now, pivotal parts have been added that make the flow more logical.

- The title has now been extended by adding a few words, but still it is not referring to the specific model.

- The data are still presented in extremely busy figures, now 7 instead of five (some of which are definitely too small to be seen properly in the Journal). These are supported by an additional eight (previously five!) also extremely busy supplementary figures. Figure 1, in fact, consists of two individual ones that have been combined for editorial reasons?

What still makes reading of this manuscript so complicated is the permanent parallel referring to the figures and at the same time to the supplementary figures even in one sentence. Even within the legends of figures and supplementary figures cross-references to each other can be found. This has been criticized before and nothing has been changed significantly. The manuscript remains very difficult to read and questions the sense of supplementary data. If one needs them to understand the main statements of the publication then they are not supplementary but necessary.

Symptomatic is that even the authors seem to be confused within the complexity of Fig 1 und Suppl. Fig 1. Thus on page 7 second half of the paragraph reference to Suppl Fig 1f is most likely wrong and should instead read S1n. And a little further down Fig 1i probably should read Fig 1l. Fig 1i is depicting the model, Fig 1 l are the data,

In summary, the results in this manuscript point to macrophages as the most important cell type in promoting repair processes in this lung injury model. IL-33 alone or together with IL-13 is necessary in this repair process and IL-33 has effects on more than one cell type involved in this process. The manuscript supports these claims in a convincing manner.

A few points (still):

- Referring to Fig 1 a and text on p 13: There is an IL-13 stimulated effect (10) and one stimulated by IL-33 (29), the combination is additive (40). The authors claim a further enhanced effect – what do they mean by that? It is additive and not synergistic.

- A bit further down the authors cite a “somewhat consistent” observation (cit 25). What does “somewhat” mean in that context? Same phrase again used on p 17 ... somewhat consistent with a report...

- P 17 end of middle paragraph the authors claim that the observed effect was specific for IL-33 since IL-13 failed to alter macrophage division in vitro. This is somewhat misleading. Wouldn't one rather check for the effect of another IL-1 family member to check for IL-33 specificity, e.g. IL-1? Given that IL-1RI is upregulated (Fig 2f) and elevated IL-1alpha (fig 1e) was measured in the lungs? Claiming specificity of a cytokine by stating that another one, that isn't even related to it, isn't effective seems a little bit arbitrary.

Reviewer #2 (Remarks to the Author):

The authors have satisfactorily addressed many of my previous concerns. However, there are still a few important points they need to address (see below Major Points). The authors performed a significant number of experiments and their findings are interesting. However, they should be careful not to over-interpret their data or make claims for mechanisms that they have not demonstrated (there are many over-interpretations in the graphical abstract).

MAJOR POINTS

1) Expression of IL-33 in club cells during homeostasis (Fig. 3a)

The authors suggest that IL-33 is released following NA injury because it is constitutively produced by club cells in the lungs of naïve mice (Fig. 3a). However, previous studies by many groups have shown that ATII cells, not club cells, are the main producers of IL-33 in naïve lungs. No evidence was found for IL-33 production in bronchial epithelium during homeostasis, as reported by several groups, including the authors of the present manuscript in their previous studies (see Fig 2C in Kearley et al., *Immunity* 2015, 42:566-579). Therefore, the authors should validate the expression of IL-33 in club cells using IL-33 reporter mice (for instance IL-33-citrine reporter) or IL-33-deficient mice;

"We respectively disagree with the reviewer on this point; using an heterozygous IL-33 gene trap (I133Wt/Gt) mice, which express the LacZreporter cassette under the endogenous I133 gene promoter, IL-33 expression was recently reported to be localized into the subsets of lung epithelial progenitors (club cells) (*J Clin Invest.* 2013 Sep;123(9):3967-82). We have added this reference and text to the revised manuscript..."

The authors should read more carefully their references before answering reviewer comments. Using in situ hybridization, Byers et al (*J Clin Invest.* 2013; 123(9):3967-82) have found NO CONSTITUTIVE EXPRESSION of I133 mRNA in club cells during homeostasis (see Fig. 5B). They detected expression of I133 mRNA in club cells only after parainfluenza virus infection. Similarly, Byers et al did not observe I133 promoter activity (visualized with the beta-gal reporter) in club cells in the absence of viral infection (see Fig. 5D). Thus, the previous observations by Byers et al cast some doubts about the specificity of the IL-33 signal in club cells in naïve mice, reported by the authors of the present manuscript in Fig. 3a.

Therefore, the authors should validate the specificity of the IL-33 staining in club cells in naïve mice using I133-deficient mice, which are widely available. In the absence of this critical control, the authors cannot claim IL-33 expression in club cells, and they need to delete the data presented in Fig. 3a from the manuscript.

2) Role of ST2 in bronchial epithelial repair

The alterations in gene expression in P2 and P3 cells isolated from ST2^{-/-} mice (Fig. 4) could be due to indirect effects caused by the absence of ST2 in the lung microenvironment. If there is a direct effect of IL-33, it is more likely to be on P2 cells and P2 to P3 differentiation rather than on P3 AAMs.

- The authors show high levels of *il1rl1* (*st2*) mRNA in P2 cells compared to P3 cells (Fig. 2e) and high levels of ST2-GFP expression in P2 cells (Fig. 2g). The authors do not show ST2-GFP expression in P3 AAMs. Why ?

They should show ST2-GFP expression on the adoptively transferred P3 AAMs if they want to call these cells ST2⁺ AAMs.

- Unfortunately, the authors have not performed adoptive transfer experiments with ST2^{-/-} P3 AAMs in ST2^{-/-} mice. In the absence of this critical negative control, it remains unclear whether the effects

on epithelial regeneration are due to ST2 signalling in the adoptively transferred AAMs or to other mechanisms. Therefore, the authors cannot claim that "ST2 function on AAMs was required for effective repair" (p10).

3) "Figure 7. Lung ILC2s were activated after NA injury and contributed to macrophage maturation through their IL-13 production."

The authors have not shown that ILC2-derived IL-13 is required for macrophage maturation. Since the authors have not performed adoptive transfer experiments using IL13-deficient ILC2s, the importance (if any) of ILC2-derived IL-13 in the NA injury model remains unknown.

A more appropriate title for Figure 7 would be "Lung ILC2s produce IL-13 after NA injury and contribute to macrophage maturation."

4) "Figure 8. Proposed cellular and molecular mechanism depicting the direct contribution of the IL-33-ST2 axis in myeloid cell differentiation/maturation as well as ILC2 activation during club cell regeneration."

There are many over-interpretations in this figure. A few examples: 1) Production of IL-33 by club cells has not yet been validated by the authors (see major point 1); 2) The authors have not shown that IL-33 is released after NA injury; 3) The authors have not shown that IL-33 acts directly on P1 cells. Actually, they show the opposite in the manuscript since the data in Fig S2C demonstrate that ST2 is not expressed on P1 cells; 4) The authors have not shown that IL-33 acts directly on P3 cells. They failed to provide the critical control (ST2^{-/-} AAMs) in the adoptive transfer experiments (see major point 2); 5) the authors have not shown that IL-13-derived from ILC2s act on P1 cells (see major point 3).

Therefore, the authors should delete Figure 8 (and graphical abstract) from the manuscript.

OTHER POINTS

5) p4, "This alarmin signals via the ST2 receptor which is expressed by several cell types in the lung, including group 2 innate lymphoid cells (ILC2s) and macrophages."

6) p5, "Three specific myeloid populations expand in the airways following naphthalene injury."

7) p5, "This macrophage expansion was associated with an early (d1 - d3) increase in BAL fluid levels of the cytokines, IL-1 α , IL-13, CCL2 and CXCL10...."

8) p6, "...in total BAL cells isolated from NA-treated mice, when compared to controls (Supplementary Fig. 1h)."

9) p6, "Interestingly, P2 and P3 subsets were positive for Ki-67 on d9 and d15 (Supplementary Fig. 1i)."

10) p6, "as evidenced by a significant reduction in total lung Scgb1a1 mRNA levels (~50% on d21 and d35, Fig. 1j), and by lower CCSP expression in lung tissue sections (Fig. 1k)."

It is necessary to move the Supplementary Fig 1n panel in Fig 1k because this is an important result.

11) p7, "Reconstitution of P3 resident macrophages completely rescued Scgb1a1 mRNA lung levels and the number of CCSP-expressing cells in macrophage-depleted mice (Fig. 1j,k)."

12) p9, "Interestingly, Il1rl1 transcripts were increased in total BAL cells and ST2⁺ monocyte-derived macrophages (P2) and resident macrophages (P3) were evident in the lung following injury, however, using ST2-GFP reporter mice, we found that levels were much higher on P2 cells (Fig. 2f,g) whereas ST2-GFP was not detected on P1 cells (Supplementary Fig. 2c)." This sentence is not clear.

13) p10, "Loss of ST2 drastically impacts the self-renewal, maturation and reparative functions of airway macrophages". Since the authors have not performed functional assays, and BMDMs rather than airway macrophages were analysed in Fig 5, a more appropriate title would be: "ST2 regulates self-renewal, differentiation and maturation of macrophages"

14) p11, "Further, absence of ST2 in P2 cells undergoing conversion into a P3 phenotype exhibited

drastic defects in..." should be corrected to "Further, P3 ST2^{-/-} cells exhibited drastic defects in..."

15) p13, "Further, ILC2-derived IL-13 has previously been shown to be required for lung regeneration following pneumonectomy⁵⁹." Reference should be 58 not 59.

16) p20, "In conclusion, this report reveals a central role IL-33-ST2 in epithelial regeneration and uncovers an important role of ST2 in promoting macrophage differentiation."

17) p44, "Figure 5. ST2 controls macrophage cell cycle progression and activation in vitro."

18) p48, "Figure S3. ST2 deficient mice exhibit impaired club cell proliferation associated with an alteration in myeloid cell function after NA injury."

19) p49, "Figure S5. ST2 deletion alters cell cycle and self-renewal of bone marrow-derived macrophages."

20) p50, "Figure S6. ST2 deletion impacts the phenotype of bone marrow-derived macrophages."

Reviewer #1 (Remarks to the Author):

The authors have addressed the points raised by this reviewer in detail. Some of them satisfactory, others not so satisfactory. In total, the manuscript is better to follow now, pivotal parts have been added that make the flow more logical.

• The title has now been extended by adding a few words, but still it is not referring to the specific model.

The titles for Nature communication manuscripts are limited to 15 non-technical words. However, the abstract informs the reader that this work was performed using the Naphthalene model.

• The data are still presented in extremely busy figures, now 7 instead of five (some of which are definitely too small to be seen properly in the Journal). These are supported by an additional eight (previously five!) also extremely busy supplementary figures. Figure 1, in fact, consists of two individual ones that have been combined for editorial reasons? What still makes reading of this manuscript so complicated is the permanent parallel referring to the figures and at the same time to the supplementary figures even in one sentence. Even within the legends of figures and supplementary figures cross-references to each other can be found. This has been criticized before and nothing has been changed significantly. The manuscript remains very difficult to read and questions the sense of supplementary data. If one needs them to understand the main statements of the publication then they are not supplementary but necessary.

As suggested by the editors, we have expanded the main figures to 9 and also significantly reduced the Supplementary data by 7 individual figures that support the various experimental methods used, flow cytometry plots, scRNA Sequencing and some additional data we previously referenced as “not shown” but were subsequently asked to include. In addition, we have removed any cross referencing to Supplementary data within the main figure legends.

Symptomatic is that even the authors seem to be confused within the complexity of Fig 1 und Suppl. Fig1. Thus, on page 7 second half of the paragraph reference to Suppl Fig 1f is most likely wrong and should instead read S1n. And a little further down Fig 1i probably should read Fig 1l. Fig 1i is depicting the model, Fig 1 l are the data.

We have corrected these typos.

In summary, the results in this manuscript point to macrophages as the most important cell type in promoting repair processes in this lung injury model. IL-33 alone or together with IL-13 is necessary in this repair process and IL-33 has effects on more than one cell type involved in this process. The manuscript supports these claims in a convincing manner.

A few points (still):

• Referring to Fig 1 a and text on p 13: There is an IL-13 stimulated effect (10) and one stimulated by IL-33 (29), the combination is additive (40). The authors claim a further enhanced effect – what do they mean by that? It is additive and not synergistic.

We believe the reviewer is referring to Fig. 6a and not Fig. 1a. Depending on the mediator examined, the effect of combined IL-13 + IL-33 stimulation on BMDM can be additive or synergistic; here the reviewer is correct in that the effect on ST2 expression on CD206+ macrophages was additive, we have amended this sentence within the results section.

• A bit further down the authors cite a “somewhat consistent” observation (cit 25). What does “somewhat” mean in that context? Same phrase again used on p 17 ... somewhat consistent with a report...

This text was previously included in response to a reviewer’s comment regarding this citation. We have now deleted this phrase/statement to provide more clarity.

• P 17 end of middle paragraph the authors claim that the observed effect was specific for IL-33 since IL-13 failed to alter macrophage division in vitro. This is somewhat misleading. Wouldn't one rather check for the effect of another IL-1 family member to check for IL-33 specificity, e.g. IL-1? Given that IL-1RI is upregulated (Fig 2f) and elevated IL-1alpha (fig 1e) was measured in the lungs? Claiming specificity of a cytokine by stating that another one, that isn't even related to it, isn't effective seems a little bit arbitrary.

The aim here was to compare the effects of IL-33 alone or in combination with IL-13 and IL-33. The role of IL-33 on monocyte-macrophage differentiation was unexpected and we were surprised to see that IL-13 had no effect under these same conditions. We have amended this statement within the discussion.

IL-1 has not previously been shown to affect AAM differentiation and activation, and the levels of IL-1 α detected in mouse lung following NA injury were extremely low compared to those of IL-13. There is one recent report showing that IL-1-derived from stromal cells acts on epithelial cells during repair, (Katsura et al, 2019 Stem Cell Reports; 12:657-666), however, most papers support a pro-inflammatory role for macrophage derived IL-1 which is opposite to the roles reported for IL-4 and IL-13 on the "M2" AAM phenotype (recently reviewed in Di Paolo & Shayakhmetov 2016, NI;17:906-913).

Reviewer #2 (Remarks to the Author):

The authors have satisfactorily addressed many of my previous concerns. However, there are still a few important points they need to address (see below Major Points). The authors performed a significant number of experiments and their findings are interesting. However, they should be careful not to overinterpret their data or make claims for mechanisms that they have not demonstrated (there are many over-interpretations in the graphical abstract).

MAJOR POINTS

1) Expression of IL-33 in club cells during homeostasis (Fig. 3a)

The authors suggest that IL-33 is released following NA injury because it is constitutively produced by club cells in the lungs of naïve mice (Fig. 3a). However, previous studies by many groups have shown that ATII cells, not club cells, are the main producers of IL-33 in naïve lungs. No evidence was found for IL-33 production in bronchial epithelium during homeostasis, as reported by several groups, including the authors of the present manuscript in their previous studies (see Fig 2C in Kearley et al., Immunity 2015, 42:566-579). Therefore, the authors should validate the expression of IL-33 in club cells using IL-33 reporter mice (for instance IL-33-citrine reporter) or IL-33-deficient mice;

"We respectively disagree with the reviewer on this point; using an heterozygous IL-33 gene trap (Il33Wt/Gt) mice, which express the LacZreporter cassette under the endogenous Il33 gene promoter, IL-33 expression was recently reported to be localized into the subsets of lung epithelial progenitors (club cells) (J Clin Invest. 2013 Sep;123(9):3967-82). We have added this reference and text to the revised manuscript..."

The authors should read more carefully their references before answering reviewer comments. Using in situ hybridization, Byers et al (J Clin Invest. 2013; 123(9):3967-82) have found NO CONSTITUTIVE EXPRESSION of Il33 mRNA in club cells during homeostasis (see Fig. 5B). They detected expression of Il33 mRNA in club cells only after parainfluenza virus infection. Similarly, Byers et al did not observe Il33 promoter activity (visualized with the beta-gal reporter) in club cells in the absence of viral infection (see Fig. 5D). Thus, the previous observations by Byers et al cast some doubts about the specificity of the IL-33 signal in club cells in naïve mice, reported by the authors of the present manuscript in Fig. 3a. Therefore, the authors should validate the specificity of the IL-33 staining in club cells in naïve mice using Il33-deficient mice, which are widely available. In the absence of this critical control, the authors cannot claim IL-33 expression in club cells, and they need to delete the data presented in Fig. 3a from the manuscript.

In our current manuscript, we have enhanced the IL-33 signal using an amplification kit and now show that a proportion of club cells express IL-33 at baseline (Fig 3a). Thus, this methodology was more sensitive and different to that used in our earlier manuscript, (Kearley et al), where we failed to detect IL-33 in naïve mouse epithelium. The polyclonal antibody used for mouse IL-33 IHC is an established reagent and has been published by many groups, further, our previous publications have indeed validated this antibody to be specific by testing in IL-33 deficient mice, these references were provided within the manuscript, see Supplemental data in reference 19.

We now provide single-cell RNA-Sequencing data of naïve mouse bronchial epithelial cells (new Fig. 4c) confirming that a proportion of Scgb1a1⁺ club cells express IL-33 at baseline. We also show that a few Scgb3a1⁺, NA-resistant club cells, also constitutively express IL-33; note that the Foxj1 positive (ciliated) cells are distinct from the IL-33⁺ cells (new Fig. 4c). Interestingly, these data also show that there is a lack of ST2 (Il1r1) and IL-IRAcP (Il1rap) mRNA expression, suggesting that epithelial cells do not respond to IL-33. Thus, this data confirms the IHC data shown in Fig 4a (old Fig. 3a) and [Redacted]

2) Role of ST2 in bronchial epithelial repair

The alterations in gene expression in P2 and P3 cells isolated from ST2^{-/-} mice (Fig. 4) could be due to indirect effects caused by the absence of ST2 in the lung microenvironment. If there is a direct effect of IL-33, it is more likely to be on P2 cells and P2 to P3 differentiation rather than on P3 AAMs.

- The authors show high levels of il1r1 (st2) mRNA in P2 cells compared to P3 cells (Fig. 2e) and high levels of ST2-GFP expression in P2 cells (Fig. 2g). The authors do not show ST2-GFP expression in P3 AAMs. Why?

- Unfortunately, the authors have not performed adoptive transfer experiments with ST2^{-/-} P3 AAMs in ST2^{-/-} mice. In the absence of this critical negative control, it remains unclear whether the effects on epithelial regeneration are due to ST2 signalling in the adoptively transferred AAMs or to other mechanisms. Therefore, the authors cannot claim that "ST2 function on AAMs was required for effective repair" (p10).

We were unable to perform successful adoptive transfer experiments with ST2^{-/-} P3 cells given it was technically challenging to retrieve enough P3 cells in the donor ST2-deficient mice. Thus, we have omitted all text referring to ST2⁺ P3 cells and/or claiming “ST2 function on AAMs was required for effective repair” throughout the manuscript, including the abstract. Further we have re-worded the adoptive transfer experiment performed in accordance with the Reviewer’s comments.

We have previously observed that the weak fluorescence emission signals from the ST2 reporter mice are not sensitive enough to detect lower levels of ST2, (in house, unpublished data), and thus, were unable to demonstrate significant levels of GFP on P3 cells at the specific time point used to examine the cells. However, we did show that ST2 mRNA and the IL-1 receptor accessory protein as well as the downstream signaling pathways of ST2 were expressed across the P1, P2 and P3 subsets at differential levels, (heatmap shown in Fig 3e), suggesting that expression of this receptor may be transient across the 3 macrophage subsets during NA-induced injury. ST2 gene expression (Fig. 3e) in the differentiation of P1 to P2 cells appears to be strongly associated with active cell-cycle progression, (i.e. pathways associated with the cell cycle are downregulated in ST2^{-/-} P2 cells, Fig 6b), and levels are significantly enhanced in differentiated P2 cells, (i.e. we can measure ST2 protein); interestingly, ST2 levels are then reduced upon differentiation into P3 cells. This hypothesis of transient ST2 receptor expression is consistent with Aoki et al, (Mol Cell Biochem, 2010; 335:75-81), who reported that ST2 expression was transient and growth dependent; this group showed increased ST2 gene expression in growing cells while levels of this receptor were significantly down-regulated when endothelial cells differentiated to form vascular structures on the extracellular membrane matrix. Further experiments with highly sensitive reporter mice would be required to track accurate receptor expression during the course of NA-induced injury and repair to confirm transient ST2 receptor expression during cellular differentiation. We have added new text within the results section to clarify the differences observed in ST2 expression across the P1-P3 macrophage subsets.

3) *“Figure 7. Lung ILC2s were activated after NA injury and contributed to macrophage maturation through their IL-13 production.” The authors have not shown that ILC2-derived IL-13 is required for macrophage maturation. Since the authors have not performed adoptive transfer experiments using IL13-deficient ILC2s, the importance (if any) of ILC2-derived IL-13 in the NA injury model remains unknown. A more appropriate title for Figure 7 would be “Lung ILC2s produce IL-13 after NA injury and contribute to macrophage maturation.”*

The title for Figure 7 has been amended as suggested by the reviewer.

ILC2-derived IL-13 has previously been shown to be required for lung regeneration following pneumonectomy, Lechner et al. Cell Stem Cell; 2017, thus, this observation was key to confirm nevertheless in a different repair model and support our central hypothesis. We have added this statement and referenced a second time to re-emphasize this point.

4) *“Figure 8. Proposed cellular and molecular mechanism depicting the direct contribution of the IL-33-ST2 axis in myeloid cell differentiation/maturation as well as ILC2 activation during club cell regeneration.”*

There are many over-interpretations in this figure. A few examples: 1) Production of IL-33 by club cells has not yet been validated by the authors (see major point 1); 2) The authors have not shown that IL-33 is released after NA injury; 3) The authors have not shown that IL-33 acts directly on P1 cells. Actually, they show the opposite in the manuscript since the data in Fig S2C demonstrate that ST2 is not expressed on P1 cells; 4) The authors have not shown that IL-33 acts directly on P3 cells. They failed to provide the critical control (ST2^{-/-} AAMs) in the adoptive transfer experiments (see major point 2); 5) the authors have not shown that IL-13-derived from ILC2s act on P1 cells (see major point 3).

Therefore, the authors should delete Figure 8 (and graphical abstract) from the manuscript.

As suggested by the reviewer, we have deleted the graphical figure.

All other points # 5) – 20) above have been amended as suggested by the reviewer.

REVIEWERS' COMMENTS:

Reviewer #2 (Remarks to the Author):

The manuscript has been greatly improved thanks to the restructuration in 9 main display items. The authors have satisfactorily addressed my previous concerns. The single cell RNAseq data of naïve mouse bronchial epithelial cells, showing IL-33 mRNA expression in a subset of club cells (Fig. 4c), are a nice addition. The authors have not been able to perform the adoptive transfer experiments with ST2^{-/-} P3 cells due to technical difficulties. Accordingly, they have modified the text of the manuscript and no longer claim that "ST2 function on AAMs was required for effective repair". Moreover, they have deleted the graphical abstract that included many over-interpretations, and have significantly toned down the statements not supported by their data. Therefore, the manuscript is now appropriate for publication.